# Single-cell profiling of trabecular meshwork identifies mitochondrial dysfunction in a glaucoma model that is protected by vitamin B3 treatment

Nicholas Tolman[1], Taibo Li[2†], Revathi Balasubramanian[1†], Guorong Li[3], Rebecca Pfeiffer[4], Violet Bupp-Chickering[1], Ruth A Kelly[3], Marina Simón[1], John Peregrin[1], Christa Montgomery[1], Bryan Jones[4], W Daniel Stamer[3*], Jiang Qian[5*], Simon WM John[1,6*]

[1]Department of Ophthalmology, Vagelos College of Physicians and Surgeons, Columbia University Irving Medical Center, New York, United States; [2]Department of Biomedical Engineering, Johns Hopkins University, Baltimore, United States; [3]Department of Ophthalmology, Duke University, Durham, United States; [4]Department of Ophthalmology, University of Pittsburgh, Pittsburgh, United States; [5]Department of Ophthalmology, Johns Hopkins School of Medicine, Baltimore, United States; [6]Zuckerman Mind Brain Behavior Institute, Columbia University, New York, United States

*For correspondence:
william.stamer@duke.edu (WDS);
jiang.qian@jhmi.edu (JQ);
sj2967@cumc.columbia.edu
(SWMJ)

†These authors contributed equally to this work.

## eLife Assessment

This study provides a **fundamental** advancement in our understanding of trabecular meshwork cell diversity and its role in eye pressure regulation and glaucoma using multimodal single-cell analysis, spatial validation, and functional testing that go beyond the current state-of-the-art. The study demonstrates that mitochondrial dysfunction, specifically in one of three distinct cell subtypes (TM3), contributes to elevated IOP in a genetic mouse model of glaucoma carrying a mutation in the transcription factor Lmx1b. While the identification of TM3 cells as metabolically specialized is **compelling**, there is somewhat limited evidence directly linking mitochondrial dysfunction to the Lmx1b mutation in TM3 cells.

**Abstract** Since the trabecular meshwork (TM) is central to intraocular pressure (IOP) regulation and glaucoma, a deeper understanding of its genomic landscape is needed. We present a multimodal, single-cell resolution analysis of mouse limbal cells (includes TM). In total, we sequenced 9,394 wild-type TM cell transcriptomes. We discovered three TM cell subtypes with characteristic signature genes validated by immunofluorescence on tissue sections and whole-mounts. The subtypes are robust, being detected in datasets for two diverse mouse strains and in independent data from two institutions. Results show compartmentalized enrichment of critical pathways in specific TM cell subtypes. Distinctive signatures include increased expression of genes responsible for (1) extracellular matrix structure and metabolism (TM1 subtype), (2) secreted ligand signaling to support Schlemm's canal cells (TM2), and (3) contractile and mitochondrial/metabolic activity (TM3). ATAC-sequencing data identified active transcription factors in TM cells, including LMX1B. Mutations in *LMX1B* cause high IOP and glaucoma. LMX1B is emerging as a key transcription factor for normal mitochondrial function, and its expression is much higher in TM3 cells than other limbal cells. To understand the role of LMX1B in TM function and glaucoma, we single-cell sequenced limbal

cells from $Lmx1b^{V265D/+}$ mutant mice (2491 TM cells). In $Lmx1b^{V265D/+}$ mice, TM3 cells were uniquely affected by pronounced mitochondrial pathway changes. Mitochondria in TM cells of $Lmx1b^{V265D/+}$ mice are swollen with a reduced cristae area, further supporting a role for mitochondrial dysfunction in the initiation of IOP elevation in these mice. Importantly, treatment with vitamin B3 (nicotinamide), which enhances mitochondrial function and metabolic resilience in other contexts, significantly protected $Lmx1b$ mutant mice from IOP elevation.

## Introduction

Aqueous humor (AH) dynamics and pressure homeostasis are important for proper ocular inflation and vision. Elevated intraocular pressure (IOP) is a key risk factor for glaucoma, a serious blinding disorder that affects 80 million people worldwide (*Quigley and Broman, 2006*). IOP is controlled by continually adjusting resistance to drainage (outflow) of AH through the trabecular meshwork (TM) and Schlemm's canal (SC), the primary route of AH outflow from the eye. AH percolates through the TM before exiting the eye through SC and then entering the venous circulation. As the most abundant constituent of the conventional outflow pathway, TM cells are central in regulating AH drainage and IOP. Despite significant advances over the past few decades, a much deeper understanding of the molecular mechanisms governing TM cell health and function, as well as the pathological alterations responsible for increased outflow resistance, elevated IOP, and increased glaucoma risk is still needed.

The TM is located at the inner aspect of the wall of the eye at the iridocorneal angle just anterior to where the iris and cornea meet (i.e. the limbus). In humans, the TM is morphologically separated into distinct zones, including the uveal, corneoscleral, and juxtacanalicular trabecular meshwork (JCT), which lies adjacent to the SC. AH first drains through the uveal and corneoscleral meshwork, whose TM cells have endothelial-like properties (*Stamer and Clark, 2017*; *Keller and Peters, 2022*; *Buffault et al., 2020*). In these regions, the TM consists of beams and plates made of various extracellular matrix (ECM) components, including fibrillar collagens (*Acott and Kelley, 2008*; *Fuchshofer et al., 2006*; *Yue, 1996*; *Vittal et al., 2005*; *Abu-Hassan et al., 2014*). The beams and plates are covered with monolayers of TM cells. AH drains through a tortuous series of intertrabecular spaces, which run between the beams, ensuring interaction with TM cells. Next, AH flows into the less structured JCT region before entering the SC. The JCT consists of TM cells that are fibroblast-like, extending their processes onto neighboring JCT and SC cells. JCT TM cells are embedded within a diffuse ECM ground substance (including glycosaminoglycans; *Stamer and Clark, 2017*; *Vranka et al., 2015*; *Keller and Acott, 2013*; *Johnson, 2006*), and have contractile smooth muscle-like properties (*Stamer and Clark, 2017*; *Tian et al., 2009*). Importantly, cells in the JCT region impose critical resistance to AH outflow with a major site of resistance to outflow being located in the tissue where the JCT and SC meet (*Acott and Kelley, 2008*; *Tamm, 2009*; *Overby et al., 2009*; *Johnson, 2006*).

Regulation of AH outflow resistance and IOP homeostasis by TM cells is complicated. AH outflow is modulated by: (1) The physical properties (shape/volume and contractility) of the TM cells. For example, some TM cells have smooth muscle-like properties that adjust outflow resistance and thus IOP by controlling cellular contractility (*Zhang and Rao, 2005*; *Thieme et al., 2000*; *Rao et al., 2005*; *Syriani et al., 2009*; *Tian and Kaufman, 2005*; *Rao et al., 2001*; *Wang et al., 2018*; *Yu et al., 2008*). (2) The effects of different TM cell types on both the composition and abundance of ECM in the JCT due to ECM synthesis/ metabolism and phagocytic activity. (3) The paracrine or physical effects of TM cells on SC and possibly the vasculature just distal to SC (*Thomson et al., 2021*; *Kizhatil et al., 2014*; *Balasubramanian et al., 2024*; *Stamer and Acott, 2012*).

Due to their importance, a more detailed molecular characterization of TM cells is still required. The degree to which specific pathways and modulatory roles are compartmentalized or shared by specific TM cell subtypes and how specific subtypes influence each other and the SC are not known. In fact, the number of molecularly distinct subtypes is currently unknown, as is the role of the >120 genes associated with elevated IOP within these subtypes (*Gharahkhani et al., 2021*). Although the TM has fewer beams and a narrower drainage path in mice, developmental, functional, and anatomical similarities between human and mouse TM make mice a valuable model for studying the molecular characteristics, heterogeneity, subtypes, and distribution of TM cells (*Chen et al., 2016*; *Smith et al., 2001*).

Single-cell resolution transcriptomic profiling is revolutionizing the molecular characterization of cell types and cellular heterogeneity (single-cell RNA sequencing, scRNA-seq; and single-nucleus, snRNA-seq). Initial single-cell-level studies have characterized cell types of the ocular anterior segment, including TM cells (*Thomson et al., 2021*; *Balasubramanian et al., 2024*; *van Zyl et al., 2020*; *van Zyl et al., 2022*; *Patel et al., 2020*)**,** but further depth, validation, and more uniform molecular consensus/naming of cell types are needed. This is especially true for TM cells. Currently, reports differ in the naming and definition of TM cell subtypes and offer limited validation by immunohistochemistry (IHC) or immunofluorescence (IF) on tissue sections or flat mounts.

Here, we more deeply characterized the mouse TM and validated findings by immunostaining tissue sections and flat mounts and by in situ hybridization (ISH). We independently isolated and sequenced limbal cells of mice at 2 institutions and characterized the same three TM cell subtypes at both. Additionally, single-nucleus assay for transposase-accessible chromatin with sequencing (snATAC-seq) identified LIM homeobox transcription factor 1 beta (LMX1B) as one of the most active transcription factors in TM cells. Variants in *LMX1B* cause IOP elevation and glaucoma (*Gharahkhani et al., 2021*; *Khawaja et al., 2018*; *MacGregor et al., 2018*; *Choquet et al., 2018*; *Gao et al., 2018*; *Shiga et al., 2018*). By analyzing *Lmx1b*$^{V265D/+}$ mutant mice (develop elevated IOP and glaucoma) (*Cross et al., 2014*; *Tolman et al., 2021*), we show that mitochondrial pathways are primarily disturbed in the TM3 cell subtype, which most strongly expresses *Lmx1b*. Treatment with nicotinamide lessened IOP elevation in *Lmx1b*$^{V265D/+}$ mutants supporting testing of therapeutics that boost mitochondrial health and function in human patients.

## Results

### Three molecularly distinct TM cell subtypes

Using scRNA-seq, we sequenced 17,914 cells from dissected limbal tissue (contains drainage structures) of 2-month-old mice. The dataset included 13,251 cells from strain C57BL/6J (B6) and 4663 cells from strain 129/Sj (129). The data for both strains were integrated (*Figure 1—figure supplement 1*). Computational analysis revealed six distinct clusters of cells that were TM-containing, epithelia, pigmented epithelia (iris and ciliary body), endothelial, immune, and neuron (*Figure 1A*). The identity of each of these cell clusters was based on various well-characterized marker genes (*Figure 1—figure supplement 2A–B*; *Thomson et al., 2021*; *Balasubramanian et al., 2024*; *van Zyl et al., 2020*). There were no major differences in the distributions of strain B6 and strain 129 cells within the relevant cell-type clusters (*Figure 1—figure supplement 3A–C*). Therefore, to improve statistical power, downstream analyses used the integrated dual-strain dataset.

Cells of cluster 1 expressed various TM genes, including *Acta2* (α-SMA), *Pitx2*, *Tfap2b*, and *Myoc* (*Figure 1—figure supplement 2A*; *Balasubramanian et al., 2024*; *van Zyl et al., 2020*; *Akula et al., 2020*; *Ostojic et al., 2008*; *de Kater et al., 1992*). Unbiased sub-clustering of Cluster 1 revealed 10 distinct sub-clusters (*Figure 1B*), which included three TM cell clusters based on signature gene expression (*Figure 1D–E*). We named these clusters TM1, TM2, and TM3. Hierarchical clustering of these cell types indicated that TM1 and TM2 are more molecularly similar to each other than to TM3 (*Figure 1C*). The remaining seven clusters were identified as cells from other ocular tissue including the iris stroma, sclera, corneal keratocytes, corneal endothelium, ciliary muscle, pericytes, and Schwann cells (*Figure 1—figure supplement 2C*; *van Zyl et al., 2022*; *Patel and Parker, 2015*; *Lopez et al., 2009*; *Liu et al., 2003*; *Toyono et al., 2015*; *Monje et al., 2018*; *Stratton et al., 2017*; *Nitzan et al., 2013*).

Next, we integrated an additional limbal tissue scRNA-seq dataset (14,912 cells, Duke University) from 3 months old C57BL/6J mice with our initial 2 months old C57BL/6J dataset. Due to batch effects most likely based on differences in cell isolation techniques and environment (Columbia University vs. Duke University, see Materials and methods), TM cells from these datasets occupied adjacent, partially overlapping, but not identical UMAP space (*Figure 1—figure supplement 4*). Individual analysis of the Duke dataset also identified three TM cell subtypes with the same marker gene expression and enriched molecular pathways compared to the Columbia University data (*Figure 1—figure supplement 4B–E*, *Supplementary file 1*). In datasets from previous mouse limbal scRNA-seq studies, there was also a high degree of overlap of marker gene expression with our three TM cell clusters (*Figure 1—figure supplement 5A–B*; *Thomson et al., 2021*; *van Zyl et al., 2020*; *Ujiie et al., 2023*).

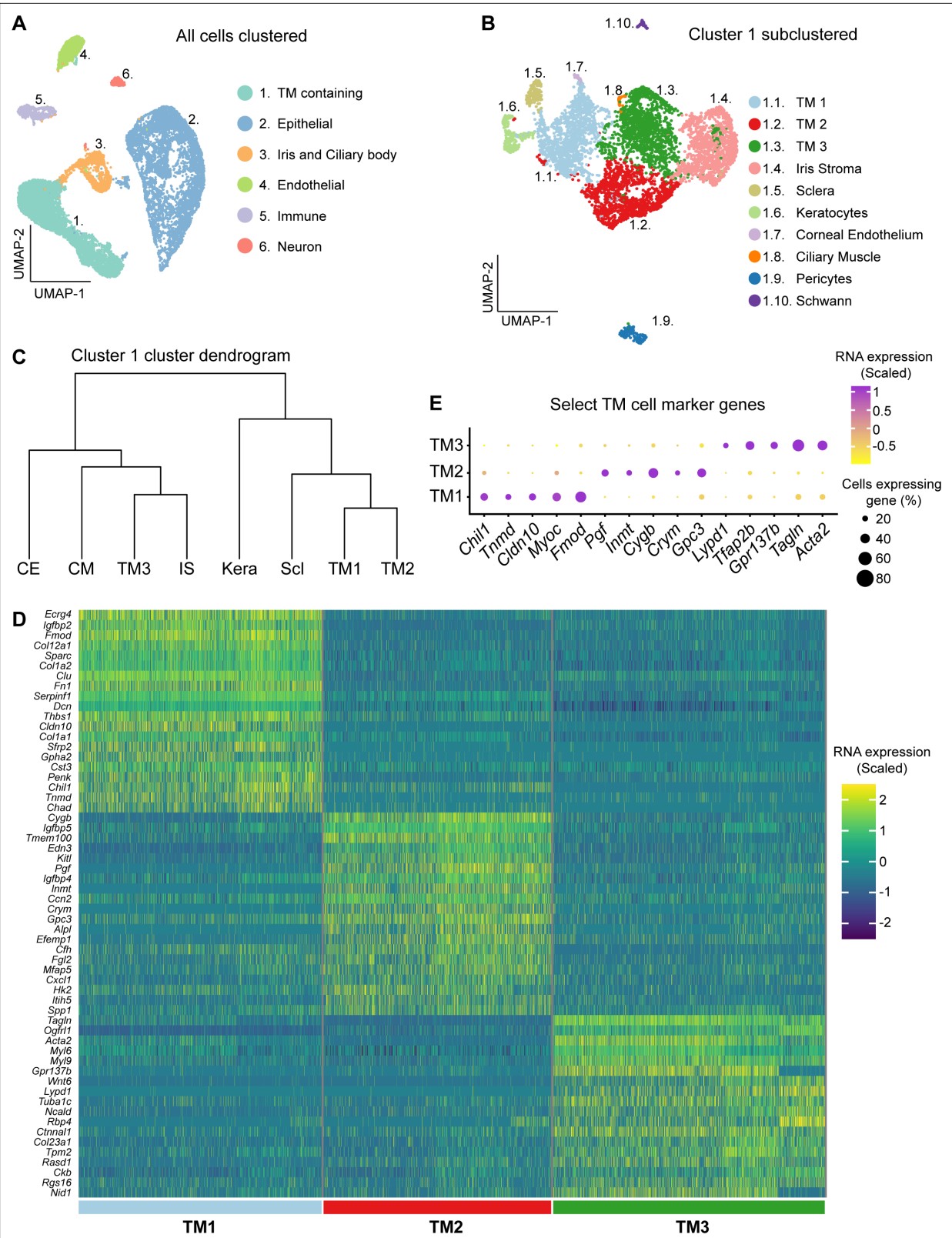

**Figure 1.** scRNA-seq data identifies three TM cell clusters. (**A**) Cells from limbal tissue dissected and sequenced at Columbia University (integrated B6 and 129/Sj datasets) are depicted in clusters on a UMAP space. (**B**) A separate UMAP representation of the trabecular meshwork (TM) containing cluster following subclustering. Three distinct TM cell clusters are identified (TM1, TM2, and TM3). (**C**) Dendrogram illustrating the relationships among these subclusters. TM1 and TM2 are more similar to each other than to TM3. (**D**) Heatmap showing the signature genes for TM1, TM2, and TM3 cells.

*Figure 1 continued on next page*

*Figure 1 continued*

The scaled RNA expression indicates how many standard deviations a given gene's expression is from the average expression across all examined cells (positive value is greater than the expression average and negative values are lower than expression average). (**E**) The scaled expression of select marker genes that robustly differentiate TM cell subsets from each other is shown with a dot plot. CE: corneal endothelium, CM: ciliary muscle, IS: iris stroma, Kera: corneal keratocytes, Scl: scleral.

The online version of this article includes the following figure supplement(s) for figure 1:

**Figure supplement 1.** Schematic of datasets and pipeline.

**Figure supplement 2.** Limbal cell cluster marker genes.

**Figure supplement 3.** Transcriptomic similarities between strain B6 and strain 129.

**Figure supplement 4.** An independent B6 scRNA-seq dataset corroborates TM cell clusters.

**Figure supplement 5.** Comparison of our TM cells to published mouse datasets.

However, some TM subtype clusters reported in previous studies expressed genes that overlapped with both of our TM2 and scleral cell clusters (*Figure 1—figure supplement 5B–C*; *Thomson et al., 2021*; *van Zyl et al., 2020*). To resolve this discrepancy, we performed IF for two of the discordantly annotated markers previously reported to be expressed in TM cells but present in our scleral cluster (CD34 and LY6C1). This immunolabeling showed that these markers are expressed in scleral but not TM cells (*Figure 2—figure supplement 1A–B*). In addition, in previous studies, cells with markers matching TM3 (e.g. Lypd1) were named uveal cells without IF confirmation (*Thomson et al., 2021*; *van Zyl et al., 2020*; *Ujiie et al., 2023*). IF revealed that our TM3 cells are neither located in the uvea nor abundant in the uveal adjacent TM but primarily reside in the anterior TM closer to the cornea (see below).

## TM subtypes differentially occupy TM zones

To further explore the sub-anatomical localization of the TM cell subtypes, we used a combination of IF and ISH to identify subtype markers. Each TM cell subtype had a higher percentage of cells expressing, and an increased average expression of, its respective subtype markers (all $p<1E-100$ enriched compared to other TM cells, *Figure 2—figure supplement 2A*). To analyze the distribution of subtypes, we first divided the TM into zones along its anterior/posterior and inner/outer axes (*Figure 2—figure supplement 2B–E*). The total area of TM occupied by each individual subtype marker was assessed in each zone on more than 150 tissue sections. This detected biases for the subtypes to be differentially localized in specific TM zones (*Figure 2A–F*). For instance, TM1 cells were significantly more frequent in the posterior and outer TM zones, while TM3 cells were overrepresented in the anterior and inner zones (all $p<0.01$). The different marker molecules that were assessed for each TM cell subtype gave highly consistent results regarding zonal occupancy (*Figure 2—figure supplement 3A–B*, *Figure 2—figure supplement 4*). In addition, we confirmed the localization biases in 3D whole mounts of the mouse limbus (*Figure 2G–H*, TM1 – MYOC, TM3 – α-SMA) (*Kizhatil et al., 2014*). Collectively, these data indicate clear localization biases that differ for the TM1 and TM3 subtypes.

To better understand the location of TM cell subtypes, we analyzed marker expression patterns in 50–60 exceptionally high-quality sections for each marker of each TM cell subtype (see Materials and methods). For this analysis, the TM was divided into eight regions (*Figure 2—figure supplement 2F*). Not only did this refine the localization bias of TM1 and TM3 cells, but it discovered that TM2 cells are most concentrated in the mid-posterior two-thirds of the TM (mid regarding inner-outer axis, all $p<0.01$, *Figure 2I–K*). Despite their biased distributions, some cells of each TM subtype were detected in most TM regions.

## Molecular comparison of TM cell subtypes

To determine the molecular functions of TM cells, we first conducted gene ontology (GO) analysis using genes that were differentially expressed genes (DEGs) in all TM cells as a group compared to other sequenced cells (*Supplementary file 2*). Additionally, we used gene set enrichment analysis (GSEA) to assess whether these GO pathways are relatively enriched or underrepresented in TM cells. Compared to other cells, TM cells were enriched for pathways associated with extracellular matrix (ECM) structure and function, including signaling associated with collagens, proteases, integrins, and

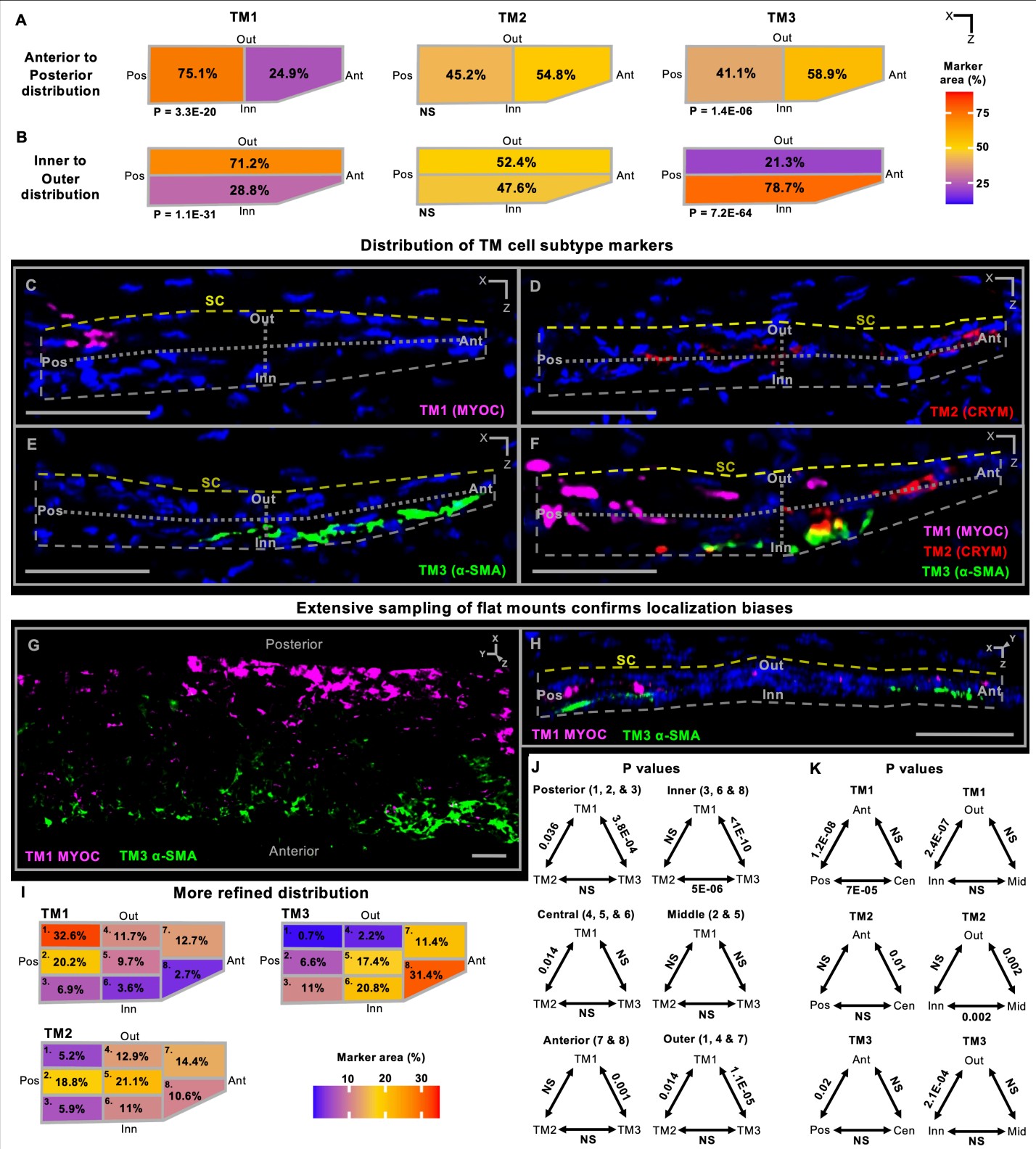

**Figure 2.** Biased localization of TM cell subtypes within the TM. (**A–B**) Summary diagrams showing the location of expression of TM1, TM2, and TM3 signature markers in the TM. For quantification of marker distributions, the TM was divided in half along both anterior-posterior (**A**) and inner-outer (**B**) axes (see *Figure 2—figure supplement 2* for localization of TM and SC). The area of the TM that was positive for each marker within each region was calculated as a percentage of the total area occupied by that marker in the entire TM on each section. The percentage of marker localization to

*Figure 2 continued on next page*

*Figure 2 continued*

each region was averaged across all analyzed sections. The average localization across all markers used for each TM cell subtype (see *Figure 2—figure supplements 3 and 4*) is shown on the diagrammatic representations of the TM using heatmaps. TM1 localization is biased towards the posterior and outer TM, whereas TM3 is biased towards the anterior and inner TM. No clear bias for TM2 was observed. A total of 286 sections were examined using immunofluorescence (IF) and in situ hybridization (ISH). Between 130 and 160 sections were examined for each TM cell subtype. (**C–F**) Representative sections for a subset of markers for each subtype used for quantification. See *Figure 2—figure supplement 1* for explanation of indicated tissue orientations. (**G**) En face image of the TM in a 3D reconstruction of a tissue whole mount, clearly demonstrating the anterior vs posterior localization bias for TM1 (posterior bias) and TM3 (anterior bias) cell types. (**H**) Orthogonal 3D image that is cropped and oriented in the same planes as the tissue sections. The inner bias for α-SMA (*Acta2*, TM3) compared to an outer bias for MYOC (TM1) is evident. (**I**) More spatially refined analyses were subsequently conducted using extremely high-quality sections (50–60 per marker) by dividing the TM into 8 smaller zones (*Figure 2—figure supplement 2*). In this more refined study, TM2 cells have a biased localization to a posterior and central region of TM (most enriched in zones 2 and 5, see *Figure 2—figure supplement 2*). (**J–K**) The TM subtype distributions were statistically compared across the combined zones (in parentheses) as indicated. ANOVA followed by Tukey's honestly significant difference test. Ant: anterior TM, Pos: posterior TM, Inn: inner TM, Out: outer TM, SC: Schlemm's canal. Gray dotted lines mark the TM zones, yellow dotted lines mark inner wall of Schlemm's canal. All scale bars: 50 µm.

The online version of this article includes the following figure supplement(s) for figure 2:

**Figure supplement 1.** Immunostaining demonstrates expression of specific marker genes in sclera but not TM.

**Figure supplement 2.** Assessing zonal distribution of TM cell subtypes.

**Figure supplement 3.** Distribution of TM cell subtypes by marker.

**Figure supplement 4.** Additional examples of TM subtype marker localization.

glycosaminoglycans (*Figure 3A*, *Figure 3—figure supplement 1A*, *Supplementary file 3*). TM cells were also enriched for growth factor signaling. Conversely, underrepresented pathways in TM cells were associated with desmosomes, peroxisomes, and certain cytoskeletal and plasma membrane elements. These pathways help define TM cells relative to other cells in the region.

Next, we compared TM cell subtypes to each other (*Figure 3B–D*). TM1 cells were most enriched for ECM pathways, particularly structural components such as collagens, integrins, and other ECM elements (*Figure 3B*, *Figure 3—figure supplement 1*). The expression of several ECM genes that are reported to impact IOP is heightened in TM1 cells, including Type VIII collagens, fibronectin, and *Ltbp2* (*Desronvil et al., 2010*; *Ali et al., 2009*; *Roberts et al., 2020*). Both TM1 and TM2 cells have enriched expression of glycosaminoglycan (GAG) genes (*Figure 3B–C*, *Figure 3—figure supplement 1B–C*). GAGs in the JCT and intertrabecular spaces form a gel-like consistency and changes in GAG abundance correlate with altered AH outflow resistance (*Knepper et al., 1981*; *Knepper and McLone, 1985*; *Knepper et al., 1996*). TM2 cells are uniquely enriched for the laminin complex, a key component of basement membranes (*Aumailley, 2013*). Pathways related to clearing debris and monitoring the extracellular environment including phagocytic vesicles, antigen presentation, and lysosomal function are also enhanced in TM2 cells. In contrast, TM3 cells are underrepresented for ECM-related pathways compared to other TM cells but are enriched for pathways related to actin binding and mitochondrial metabolism (*Figure 3D*, *Figure 3—figure supplement 1D*). The actomyosin system plays a crucial role in maintaining TM cell contractility and in regulating outflow facility (*Zhang and Rao, 2005*; *Thieme et al., 2000*; *Rao et al., 2005*; *Syriani et al., 2009*; *Tian and Kaufman, 2005*). The enrichment of mitochondrial metabolism pathways in TM3 cells emphasizes their energetic needs and their expected heightened susceptibility to genetic and environmental contexts that compromise metabolic functions. Importantly, genetic variation in mitochondrial/metabolic pathways contributes risk for IOP elevation and glaucoma (*Aboobakar et al., 2023*; *Zhang et al., 2023*; *Xin et al., 2022*; *Khawaja et al., 2016*).

We next compared growth factor signaling and ligand-target interactions between TM cell subtypes and other cells. TM1 cells were enriched for transforming growth factor beta (TGF-β) pathway signaling (*Figure 3B*). Receptor ligand activity was enriched in TM2 cells while the membrane raft pathway (a site of signal transduction) was enriched in TM3 cells (*Figure 3C–D*). Next, we used LRLoop (*Xin et al., 2022*), which was developed based on NicheNet (*Browaeys et al., 2020*), to predict significant ligand-target interactions between TM cells and other local endothelial cells that are relevant to AH drainage (*Figure 4A–D*). TGF-β ligands (*Tgfb1*, 2, and 3) were predominantly expressed in TM1 cells. By contrast, TM2 cells expressed molecules critical for vascular function, such as *Angpt1*, *Vegfa*, and *Edn3* (*Figure 4—figure supplement 1A–D*; *Shibuya, 2011*; *Saharinen et al., 2017*; *Genovesi et al.,*

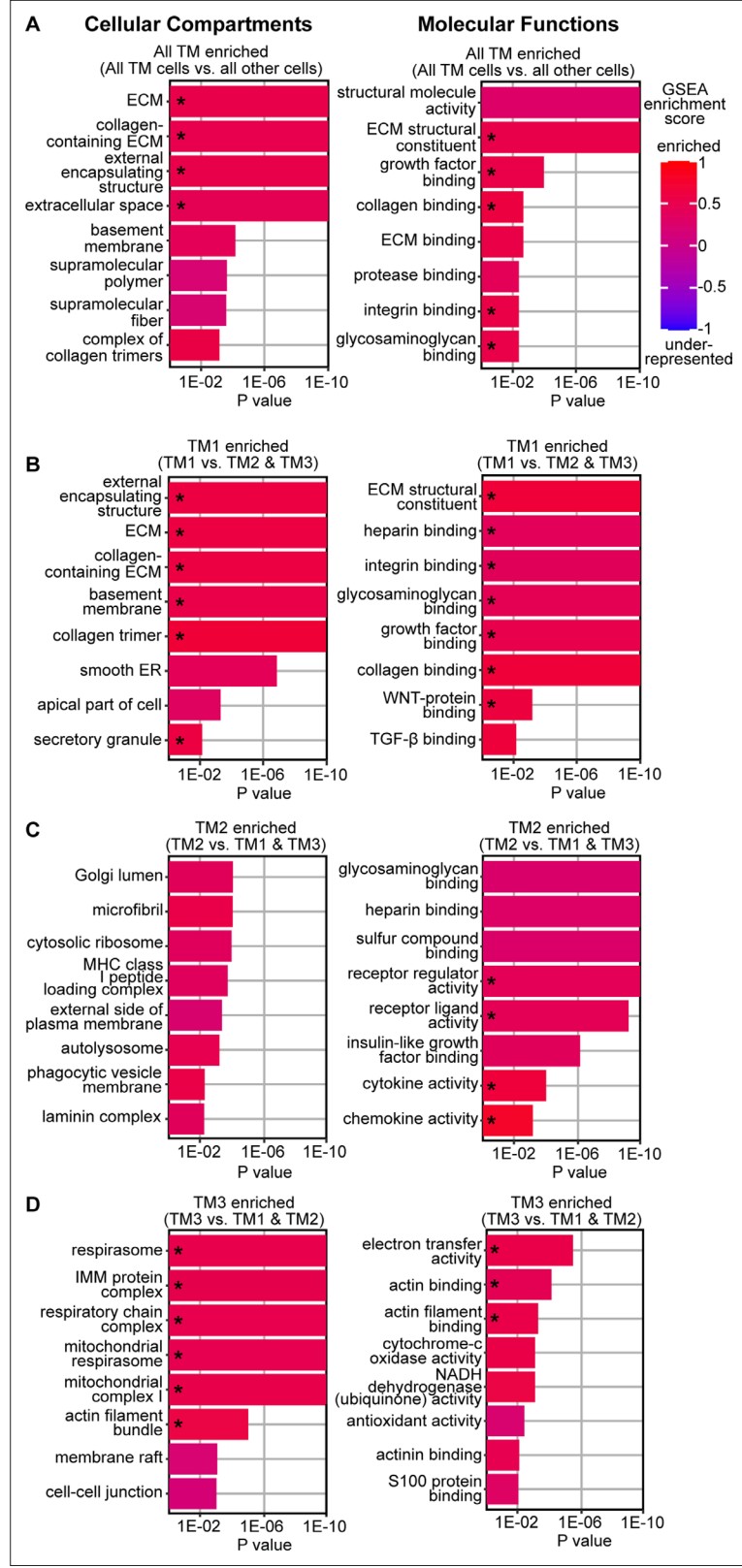

**Figure 3.** Pathway analysis suggests differential involvement of TM cell subtypes in extracellular matrix regulation, growth factor signaling, and actin-binding. (**A**) Molecular comparison of all three TM cell subtypes to all other sequenced cells using gene ontology (GO) of differentially expressed genes (DEGs). The top five most significant pathways as well as three additional pathways of interest are shown for each indicated comparison (adjusted p

*Figure 3 continued on next page*

*Figure 3 continued*

values, X-axis was cutoff at p<1E-10). GSEA analysis was also used to assess the enrichment or underrepresentation of these GO pathways with GSEA scores color coded on the GO charts above. Pathways significantly different by GSEA analysis are indicated by asterisks. Overall, TM cells have over-representation of various extracellular matrix (ECM) molecules/ pathways including collagens, glycosaminoglycans, and integrins compared to non-TM cells. Growth factor signaling is also enriched in TM cells compared to other cell types. (**B**) When comparing TM cell subtypes, TM1 cells are further enriched for ECM molecules such as collagens, glycosaminoglycans, and integrins as well as TGF-β signaling. (**C**) TM2-enriched ECM pathways include glycosaminoglycan binding and the laminin complex. TM2 cells are also enriched for receptor-ligand signaling and insulin growth factor binding. (**D**) Actin-binding and mitochondrial metabolism genes are enriched in TM3 cells. ECM = extracellular matrix. ECS = extracellular structure. ER = endoplasmic reticulum. IMM = inner mitochondrial membrane.

The online version of this article includes the following figure supplement(s) for figure 3:

**Figure supplement 1.** TM cell pathway analysis.

**Figure supplement 2.** Mouse TM subtype marker expression in human TM cells.

**Figure supplement 3.** Nuclear morphology of each TM subtype.

**Figure supplement 4.** GWAS gene expression in limbal cells.

---

*2022*). These TM-secreted molecules have been directly associated with SC maintenance and IOP (*Thomson et al., 2020*; *Fujimoto et al., 2016*; *Reina-Torres et al., 2017*; *Comes and Borrás, 2009*).

## Comparison to human TM cells

We next worked to relate our findings to human data. We compared marker genes for each of our TM cell subtypes to sequenced human TM cells from two previous studies (*van Zyl et al., 2020*; *Patel et al., 2020*). These previous studies differed in the number of human TM cell subtypes that they reported and in the identities of their clusters. van Zyl et al. identified three TM cell subtypes that they called JCT, beam A, and beam B (*van Zyl et al., 2020*), whereas Patel et al. identified two subtypes that they called fibroblast-like and myofibroblast-like (*Patel et al., 2020*). TM1 marker genes were most enriched in van Zyl et al's human JCT cells (*Figure 3—figure supplement 2A*). Similarly, *Dcn* was expressed in mouse TM1 cells and was localized to fibroblast-like TM cells adjacent to SC by Patel et al. TM1 cell nuclei were typically shorter with greater sphericity than the elongated endothelial-like nuclei of TM3 cells (*Figure 3—figure supplement 3A–D*). This nuclear morphology was more consistent with a JCT than beam cell identity. Together, these data support a JCT identity for TM1 cells. TM2 cells were more closely related to human beam A and beam B cells than to the JCT cells reported by van Zyl et al. (*Figure 3—figure supplement 2B*). TM2 cells also shared expression of the *RSPO2* marker gene with the human myofibroblast-like cells reported by Patel et al. Together, this suggests that TM2 cells are myofibroblast-like beam cells. TM3 cells also shared marker genes with Patel et al's myofibroblast-like cells (*TAGLN* and *ACTA2*), suggesting TM3 cells are also beam cells. There was relatively equal TM3 marker gene enrichment across all human TM cell subtypes of van Zyl et al. (*Figure 3—figure supplement 2C*). Nuclei immediately adjacent to TM2 and TM3 cell marker staining had an elongated, endothelial-like shape (*Figure 3—figure supplement 3A–C*). This nuclear morphology was also consistent with the location of TM2 and TM3 cells on beams and analogous to the endothelial-like morphology of human beam cells (*Stamer and Clark, 2017*).

Next, we evaluated the expression of genes implicated in elevated IOP and glaucoma by genome-wide association studies (GWAS). TM1 and TM2 cells had similar expression levels of GWAS genes associated with IOP elevation and primary open-angle glaucoma (POAG), respectively (*Figure 3—figure supplement 4A*). TM3 cells had slightly less expression of these GWAS genes compared to TM1 and TM2. To better understand the genes driving GWAS gene enrichment in different TM subtypes, we studied pathways that are significantly enriched for GWAS genes (*Figure 3—figure supplement 4B–C*, see *Supplementary file 4* for complete list). TM1 cells were the most enriched for genes involved in ECM organization, cell-substrate adhesion, and growth factor stimulus (*Figure 3—figure supplement 4D*). All 3 TM subtypes had similar expression of GWAS genes associated with hormone stimulus, *Vegf* signaling, and *Rho* kinase signaling. TM3 cells expressed lipid and carbohydrate mitochondrial metabolism pathway genes at higher expression levels and in more cells than did TM1 and TM2 cells (*Figure 3—figure supplement 4E*). The same was true for actin binding-associated GWAS

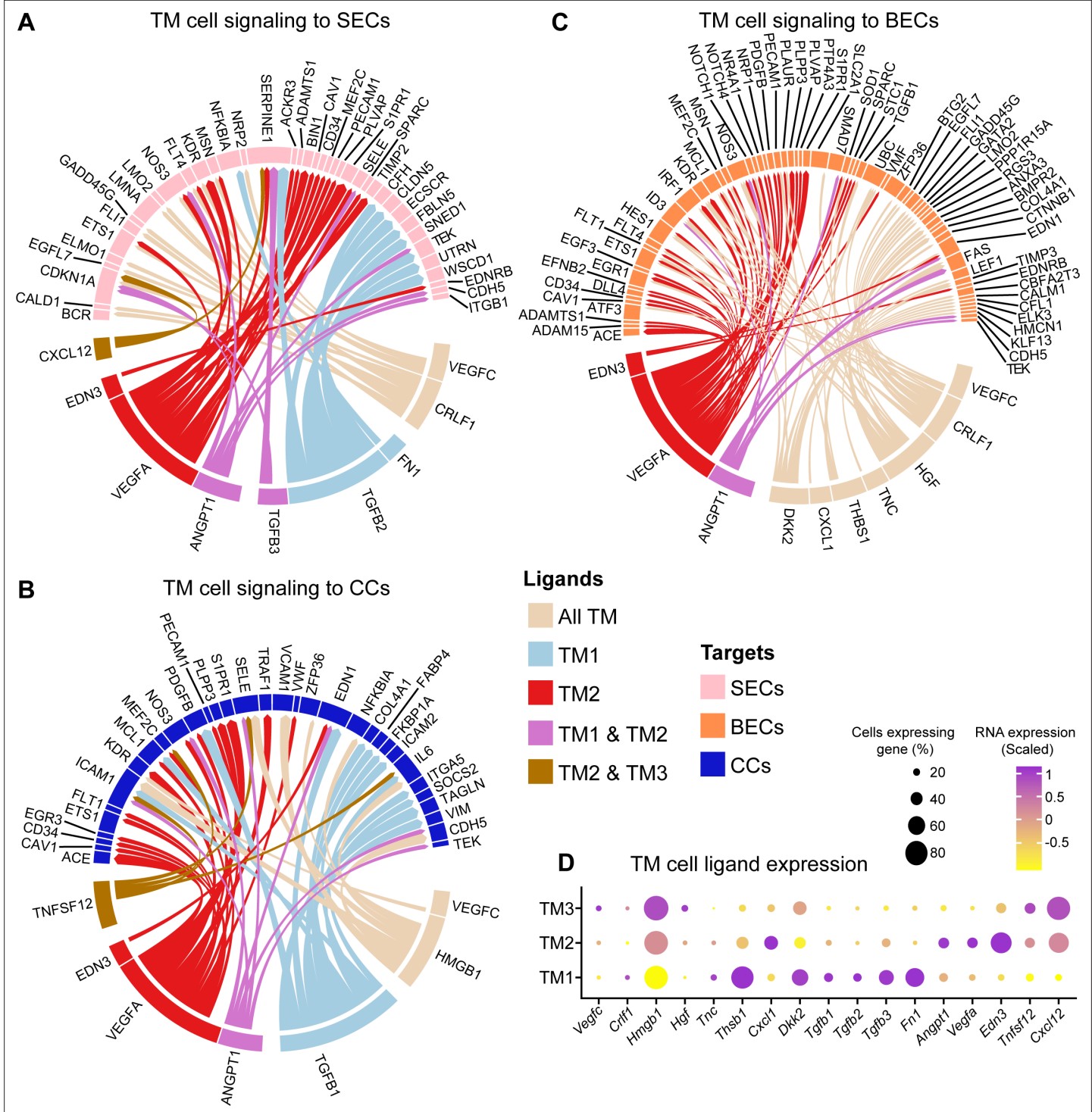

**Figure 4.** Signaling interactions directed from TM cells to Schlemm's canal and vascular endothelial cells differ across TM subtypes. (**A**) Circos plot showing the top predicted interactions between TM cell ligands and Schlemm's canal endothelial (SEC) target molecules. The top predicted targets in all TM cells are split by their expression across subtypes. TM1 and TM2 participate in more predicted signaling events compared to TM3. TM1-biased ligands include members of the *Tgf-β* and fibronectin signaling families. Various endothelial trophic factors are predominantly expressed in TM2, including endothelin 3, vascular endothelial growth factor A, and angiopoietin 1. See **Supplementary file 4** for a comprehensive list of ligand-target interaction data. (**B–C**) TM cell ligands that signal to collector channel (CC) endothelial cells (**B**) or blood endothelial cells (BECs, **C**). (**D**) The expression of all ligands predicted to have signaling interactions with SECs, CCs, or BECs is displayed across TM cell subtypes (Dot plot).

The online version of this article includes the following figure supplement(s) for figure 4:

**Figure supplement 1.** TM-TM and SEC-TM signaling.

genes in TM3 cells versus TM1 and TM2. Genetic variation in lipid and carbohydrate mitochondrial metabolism pathways is associated with IOP elevation and glaucoma (*Khawaja et al., 2016*).

## Regulation of TM cell gene expression

To better understand the overlap and differences in transcriptional control of TM cell gene expression, we performed single-cell resolution multiome sequencing. Using limbal strip tissue isolated from a separate cohort of mice than those used for scRNA-seq, we simultaneously performed both single nucleus RNA sequencing (snRNA-seq) and snATAC-seq. We thus captured both gene expression and open chromatin data throughout the genome in the same cell. Using the snRNA-seq data, we clustered cells into various cell types. The generated clusters and their marker genes generally agreed with our scRNA-seq analyses (*Figure 5A–B*, *Figure 5—figure supplement 1A*). Notably, there was a significant correlation between gene expression (RNA-based clustering) and chromatin accessibility (ATAC-based clustering) with adjusted Rand index of 0.20 (p<0.001, permutation test, *Figure 5—figure supplement 1B–D*). This included an increase in chromatin accessibility at the promoters of marker genes in their respective TM cell subtypes (*Figure 5C–E*). To discover the key transcriptional regulators within TM cells, we examined the correlation between the expression of each transcription factor (TF) and their chromatin accessibility at predicted binding sites within all TM cells. Various TFs that either activated or repressed transcriptional activity exhibited good correlations with gene expression across TM cells (*Figure 5F*). Such activating TFs include *Tcf21*, *Arid3b*, and *Myc*, while repressing TFs include *Meox2*, *Junb*, and *Sox6*. Notably, *LMX1B*, an important gene in glaucoma GWAS (*Gharahkhani et al., 2021*; *Khawaja et al., 2018*; *MacGregor et al., 2018*; *Choquet et al., 2018*; *Gao et al., 2018*; *Shiga et al., 2018*), is among the most strongly activating TFs in TM cells.

## Metabolism-supporting treatment lessens IOP elevation in *Lmx1b^{V265D}* mice

To evaluate the utility of our new TM cell atlas, we used it to examine how *Lmx1b* mutations affect the TM cell transcriptome and to identify potential mechanisms underlying IOP elevation. We selected LMX1B because it causes IOP elevation and glaucoma in humans and was identified as a highly active transcription factor in our TM cell dataset. To do this, we analyzed scRNA-seq data generated for limbal cells from mice with a dominant mutation in *Lmx1b* (*Lmx1b^{V265D/+}*), which included 2491 *V265D* mutant TM cells. These mutant C57BL/6J mice develop early onset IOP elevation (*Cross et al., 2014*; *Tolman et al., 2021*; *Zhang et al., 2024*). We compared gene expression in *V265D* mutant TM cells after integration with our wild-type data (all data B6 background and postnatal day 60, *Figure 6A*, *Figure 6—figure supplement 1A*). Results show that *Lmx1b* is minimally expressed in TM1 and TM2 but is highly expressed in TM3 cells (*Figure 6B*). Next, we analyzed DEGs and pathway differences between WT and *V265D/+* cells across each TM cell subtype. Across all TM subtypes, various genes related to ECM function and growth factor signaling were downregulated in mutants, including glycosaminoglycans and insulin growth factor binding proteins (*Figure 6C*, see *Supplementary file 6* for complete list of pathways). Pathways enriched in mutant TM cells were associated with ribosomes and calcium signaling. We then examined the differential responses to the *Lmx1b* mutation by analyzing pathways dysregulated in each TM subtype that were not common across all TM cells (*Figure 6D–F*, *Figure 6—figure supplement 1B*). *V265D* mutant TM1 cells had a downregulation of collagen synthesis/metabolism gene expression relative to WT cells, particularly in the production and turnover of fibrillar collagens. TM2 cells with *Lmx1b* mutations had upregulation of immune signaling pathways. The highly *Lmx1b*-expressing TM3 cells exhibited perturbed mitochondrial metabolism, with a downregulation of genes in complex I and ATP synthase. In addition, mutant TM3 cells showed an upregulation of protein tagging genes. However, there was a downregulation of the polyubiquitin precursor gene (*Ubb*, p=4.5E-30), indicating a general dysregulation of pathways that tag proteins for degradation. These results document altered mitochondrial function and proteostasis in TM3 cells. Although the documented gene expression changes strongly suggest metabolic and mitochondrial dysfunction, they do not directly prove it. Using electron microscopy to directly evaluate mitochondria in the TM, we found a reduction in total mitochondria number per cell in mutants (p=0.015, *Figure 6G*). In addition, mitochondria in mutants had increased area and reduced cristae (inner membrane folds), consistent with mitochondrial swelling and metabolic dysfunction (all p<0.001 compared to WT, *Figure 6G–H*).

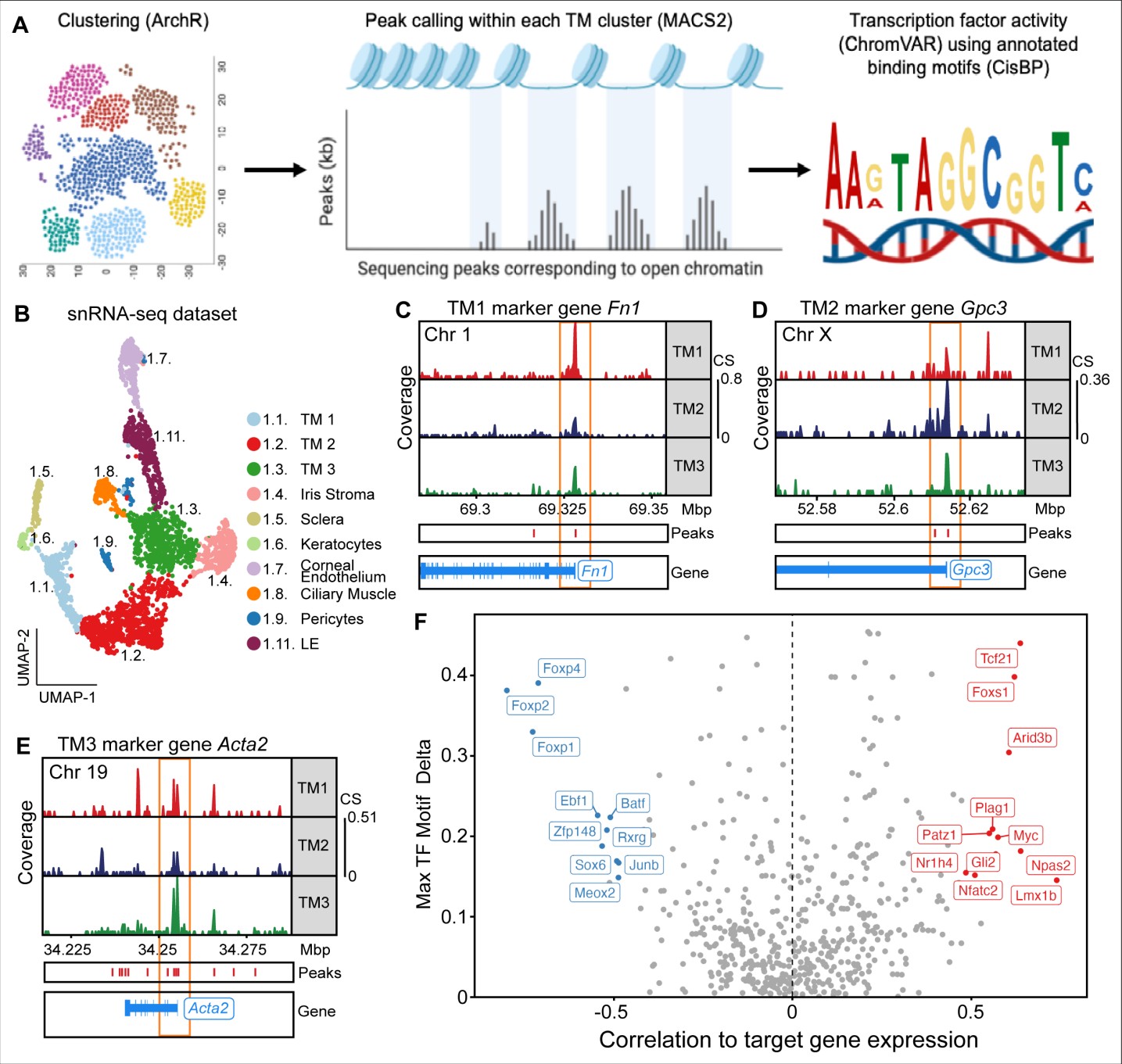

**Figure 5.** Transcription factors dictating gene expression in TM cells. (**A**) A schematic representation of the pipeline used to identify open chromatin sites and active transcription factors (TFs). Individual cells were profiled using both single nucleus (sn) RNA and ATAC sequencing (multiome). TM cell clusters were identified using the snRNA-seq data, while significantly open chromatin regions were identified using the snATAC-seq data. Active TFs were determined based on the odds ratio of TF binding motifs within these open chromatin regions. Schematic created with BioRender.com (see here, here, and here). (**B**) UMAP of subclusters derived from TM cell containing cluster (cluster 1) in the snRNA-seq data. (**C–E**) Example ATAC tracts for the promotor regions of selected marker genes for TM1 (**C**), TM2 (**D**), and TM3 (**E**). Each of these marker genes has greater promoter accessibility in the TM cell subtype in which its RNA expression is enriched (orange box). The aligned marker gene positions are shown. Coverage = normalized ATAC signal reads in transcription start site. CS = coverage scale. (**F**) Correlations between RNA expression levels (snRNA-seq dataset) of each TF and the chromatin accessibility levels (snATAC-seq dataset) of their respective predicted target binding motifs across all TM cell subtypes (see Materials and methods). Select TFs with strong positive (red) or negative correlations (blue) are named.

The online version of this article includes the following figure supplement(s) for figure 5:

**Figure supplement 1.** Analysis of ATAC dataset.

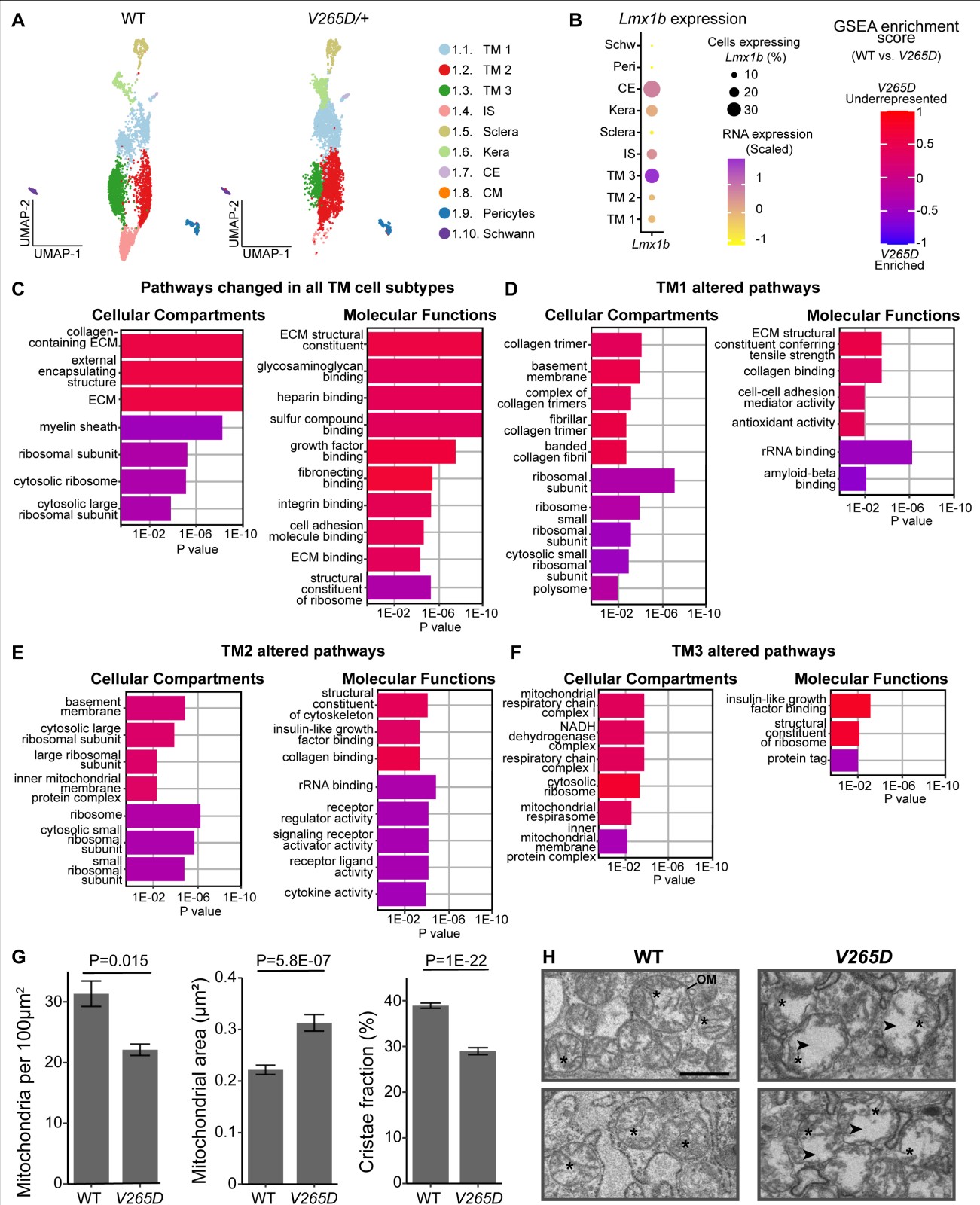

**Figure 6.** Differential molecular responses of TM cell subtypes to an *Lmx1b* mutation. (**A**) UMAPs of limbal cells from WT and *Lmx1b*[V265D/+] (*V265D*) mutant mice (both genotypes B6 background). (**B**) *Lmx1b* expression across limbal cell types. *Lmx1b* expression is highest in TM3 cells. (**C**) Pathways that are changed in all TM cell subtypes. (**D–F**) Top pathways that are significantly altered when comparing each indicated TM cell subtype across genotypes. Many of these pathways were not significantly changed when comparing all TM cell subtypes as a group, indicating that the *V265D*-induced changes

*Figure 6 continued on next page*

*Figure 6 continued*

are strongest within each TM cell subtype. *V265D* has a pronounced effect on mitochondrial pathways in TM3 cells but much less so in other TM cells. ECM = extracellular matrix. ECS = extracellular structure. (**G**) Mitochondria in *V265D* mutant TM cells are reduced in number per cell, have increased area, and decreased cristae area (cristae fraction is ratio of area occupied by cristae: total area of mitochondria; shown as bar plots with standard error). A total of 160–180 mitochondria were analyzed across 3 eyes per genotype (approximately 60 mitochondria per eye). Groups were compared using Student's t-test. (**H**) Representative images show mitochondria defined by their outer membranes (OM) and inner membrane folds (cristae). Asterisks indicate a subset of representative cristae for each genotype. Mutant mitochondria exhibit increased area (space enclosed by the OM) and disorganized cristae structure. Arrows highlight regions in mutant mitochondria that lack cristae. Scale bar = 500 nm.

The online version of this article includes the following figure supplement(s) for figure 6:

**Figure supplement 1.** Further pathway analyses of *Lmx1b^V265D^* vs WT.

Our findings most strongly implicate perturbed metabolism within TM3 cells as responsible for IOP elevation in an *Lmx1b* glaucoma model. Nicotinamide (NAM) is known to boost nicotinamide adenine dinucleotide (NAD), supporting healthy metabolism/ mitochondrial function (*Schöndorf et al., 2018*; *Williams et al., 2017b*). Thus, we hypothesized that the metabolic abnormalities in TM3 cells underlie the IOP elevation that causes glaucoma, and that NAM treatment may protect the

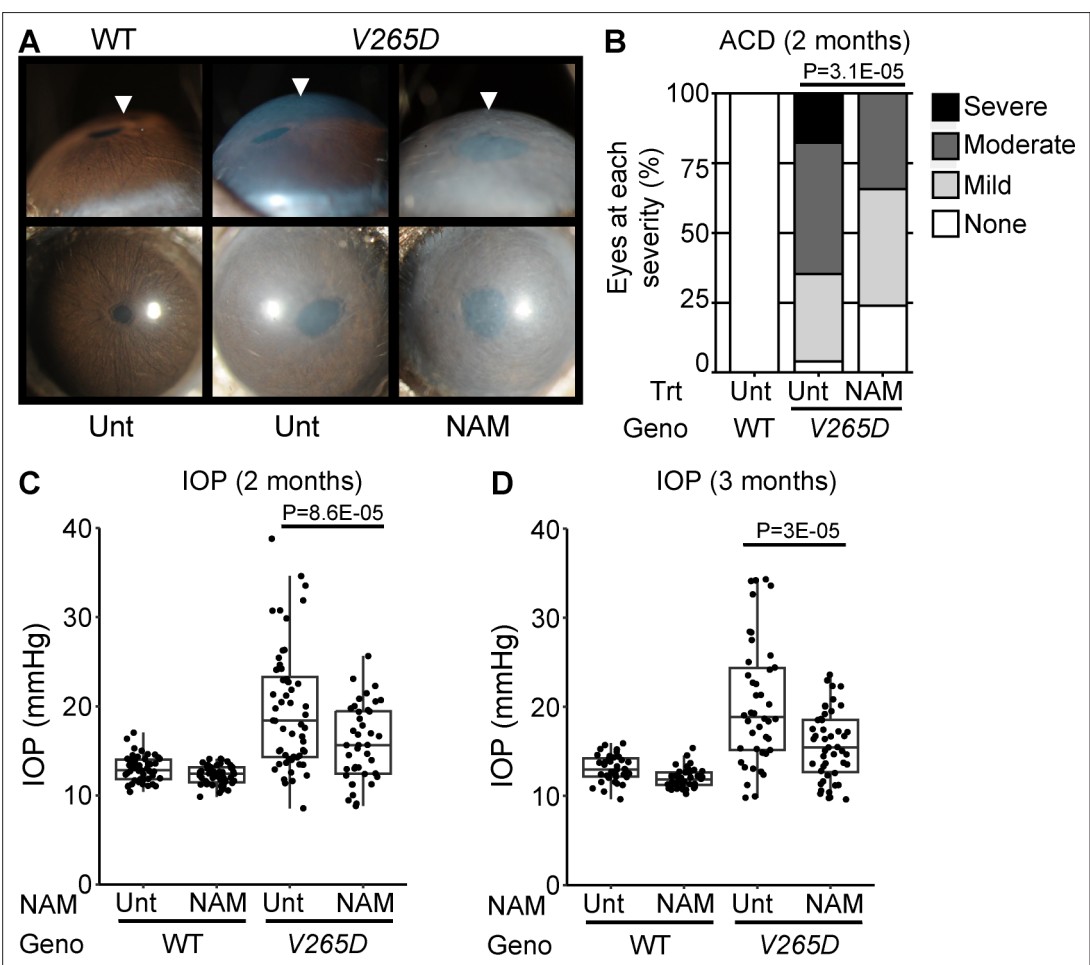

**Figure 7.** Nicotinamide treatment protects from IOP elevation in *Lmx1b* mutants. (**A**) Representative photos of eyes from mice of the indicated genotypes and treatments (UNT = untreated, NAM = nicotinamide treated, 550 mg/kg/day provided in drinking water). NAM treatment substantially lessens anterior chamber deepening (ACD), a sensitive indicator of IOP elevation in mice. The WT and NAM-treated mutant eyes have very shallow anterior chambers, while the untreated mutant eye has an obviously deepened chamber (arrowheads). (**B**) Distributions of anterior chamber depth based on previously defined scoring system (*Tolman et al., 2021*). Groups compared by Fisher's exact test. (**C–D**) Boxplots of IOP (interquartile range and median line) in WT and mutant eyes of both NAM-treated and untreated groups. NAM treatment significantly lessens IOP elevation in mutants compared to untreated mutant controls. Groups compared by ANOVA followed by Tukey's honestly significant difference (see **Supplementary file 7** for additional statistics). n≥35 eyes examined in each WT group and n≥40 in *V265D* mutant groups. Geno = genotype.

*Lmx1b* mutant TM. Beyond *LMX1B*, this hypothesis is relevant more commonly to POAG treatment because metabolism-relevant genes are implicated by GWAS (and their expression is enriched in TM3 cells, *Figure 3—figure supplement 4E*). To test our hypothesis, we treated mice by adding NAM to their drinking water starting at postnatal day 2 and continuing into adulthood. Results showed that NAM supplementation significantly protected against anterior chamber deepening (ACD, a symptom of IOP elevation, *Figure 7A–B*) and IOP elevation (*Figure 7C–D*). These results support NAM as a treatment to prevent TM damage and IOP elevation in LMX1B-mediated glaucoma with potential general relevance to POAG.

## Discussion

This study provides a comprehensive multimodal analysis of mouse limbal cells that focuses on the TM. We defined three TM cell subtypes with unique marker panels that can be used for their identification across different strains, laboratories, and processing methods. We validated key marker gene expression by IF and ISH, discovering biases for specific TM subtypes to reside in different parts of the TM. For example, TM1 cells have a location bias for the outer, posterior TM, while TM3 cells are more abundant in the inner, anterior TM. Importantly, our pathway analyses suggest that the different TM subtypes have overlapping roles. All three TM subtypes are enriched in ECM synthesis and ECM metabolism compared to other limbal subtypes. However, there is also compartmentalized enrichment of some processes to specific subtypes. For example, ECM production and turnover, secreted ligand signaling, debris removal, immune surveillance, actin-associated cell contraction, and increased metabolic gene activity (as discussed below) are enriched in specific cell subtypes. Using ATAC-seq data, we discovered key TFs controlling gene expression in TM cells, including LMX1B. Importantly, we implicate LMX1B in metabolic control in TM cells. More specifically, we demonstrate that LMX1B is the most active TM cell TF and is enriched in TM3 cells, while mutation of *Lmx1b* induces prominent mitochondrial defects in TM3 cells. Finally, we show that therapeutically targeting TM3 dysfunction with a mitochondrial/metabolism supporting nutrient (nicotinamide) protects from IOP elevation in the *Lmx1b* glaucoma model. We also suggest that a TM3-like cell is generally relevant to metabolic pathways in human glaucoma based on enriched expression of human glaucoma GWAS gene in TM3 cells. Together, these data provide a wealth of new molecular information about TM cells and suggest a new approach for preventing IOP elevation.

### Mouse trabecular meshwork subtype classification

Here, we provide evidence for three TM cell subtypes in mice that are robust across datasets from different institutions, using different tissue processing and sequencing protocols. Although others (*Thomson et al., 2021*; *Ujiie et al., 2023*) subdivided TM cell subtypes beyond the three defined in the present study, at this point, we decided against further subdivision (See *Figure 1—figure supplement 5*) for the following reasons. We lacked confidence in the higher resolution clustering, as it was not consistent across datasets, did not produce unique marker gene sets, and could have resulted in over-clustering due to stochastic or stress-induced differences between subpopulations of cells. Moreover, the number of reported mouse TM cell clusters in previous studies was inconsistent. Although three TM subtypes were reported by van Zyl et al. and Ujiie et al. (*van Zyl et al., 2020*; *Ujiie et al., 2023*), the molecular features of cells in these clusters were not the same across these studies. Differences between studies may be due to batch effects, different computational methods, and possibly over-clustering. Most importantly, there was very limited validation by IF, IHC, or other methods in previous studies. Only the JCT type of TM cell in these previous studies was assessed by means other than scRNA-seq (IF and ISH; *van Zyl et al., 2020*). Additionally, Thompson et al. treated cells with Y27632, a ROCK inhibitor, which could alter TM cell transcriptomes. It is also important to point out that both Thompson et al. and van Zyl et al. used albino mouse strains, which can impact TM cell development and function (*Libby et al., 2003*). Finally, Ujiie et al. sequenced TM cells at 3–4 weeks, which is prior to full maturation of the TM structure (*Smith et al., 2001*), while our ongoing developmental studies show substantial molecular maturation of TM cells between P21 and P60.

As van Zyl et al.'s data is deposited in the Broad Institute's single-cell portal, we further compared their data with ours. Although both van Zyl et al. and our current paper identified three TM subtypes, our work does not fully agree. Some of the cells named beam A and beam Y by van Zyl et al. express

markers present in our TM2 cell subtype, while others express markers that are absent in TM2 cells but present in our scleral cells. Our IF studies confirm that these discordant markers are not expressed in TM cells, but instead show that they are in fact scleral cells. These scleral cells highly express fibroblast markers including *Mfap5*, *Clec3b*, and *Tnxb*, suggesting they are scleral fibroblasts. Cells initially named uveal cells by van Zyl et al. (and subsequently by Ujiie et al. and Thompson et al.) express markers of our TM3 cells. Consistent with our gene expression data, our IF shows that these van Zyl markers are expressed by cells that are primarily located in the TM region that is closer to the cornea and not in the uvea or the TM region adjacent to the uvea. Thus, our current study resolves some previous inconsistencies in classifying mouse TM cell subtypes. It lays a firmer foundation for continued understanding of TM cell types and their biology. Moving ahead, sequencing cells at greater depths will enhance TM cell characterization and improve comparisons between datasets.

## Differential roles of specific mouse TM cell subtypes and comparison to human

As a group, all TM cells share various molecular properties that distinguish them from other ocular cells. Our analyses determine the expression of IOP and glaucoma genes identified by GWAS in TM cell subtypes as well as in other limbal cells. Pathways enriched across all TM cell subtypes based on RNA-seq are largely related to ECM (including collagens, integrins, and glycosaminoglycan pathways), cell-cell signaling, and growth factor signaling. This fits with an essential role of the TM in both ECM synthesis/turnover and in signaling to the SC to maintain IOP (*Thomson et al., 2021*; *Balasubramanian et al., 2024*; *Stamer and Acott, 2012*; *Keller et al., 2009*; *Fuchshofer and Tamm, 2009*). In addition, the expression of genes that we document generally agrees with the literature. For example, the following genes and signaling molecules have been reported in TM cells: WNT signaling (*Wang et al., 2008*), TGF-β signaling (*McDowell et al., 2013*; *Shepard et al., 2010*; *Rudzitis et al., 2025*; *Fleenor et al., 2006*; *Gottanka et al., 2004*; *Mody et al., 2021*; *Wordinger et al., 2007*), integrin binding (*Gagen et al., 2014*; *Faralli et al., 2019*; *Yang et al., 2022*), actin cytoskeletal networks (*Bermudez et al., 2017*), calcification genes (*Xue et al., 2007*; *Borrás and Comes, 2009*), and Myocillin (*Borrás and Comes, 2009*; *Saccuzzo et al., 2023*; *Fingert et al., 2002*; *Sharma and Grover, 2021*). Additionally, genes associated with intermediate filaments, which function in determining cell shape and structure and can also act as mechanical stress absorbers (*Herrmann et al., 2007*), are underrepresented in TM cells. Lower levels of intermediate filaments may enable TM cells to alter their shape to respond to changes in IOP.

Mouse TM1 cells closely resemble sequenced human JCT cells. Like human JCT cells, the majority of TM1 cells are located nearest to the SC inner wall, particularly adjacent to the posterior of the mouse SC. Human JCT cells are generally accepted to be embedded in a more diffuse ECM and to not cover the collagenous trabecular beams (*Stamer and Clark, 2017*; *Vranka et al., 2015*; *Keller and Acott, 2013*). In vivo, a proportion of TM1 cells have a shorter, more spherical nuclear morphology, consistent with JCT cell identity. Importantly, the JCT region is critical in determining resistance to AH outflow and in regulating IOP (*Acott and Kelley, 2008*; *Johnson, 2006*; *Stamer and Acott, 2012*; *Ethier, 2002*). JCT cells have fibroblast-like properties, including the secretion of ECM proteins and degradation enzymes to support continuous ECM remodeling (*Stamer and Clark, 2017*; *Keller et al., 2009*; *Acott et al., 1988*). The subset of TM1 cells that lie distant from SC has a more beam cell-like, elongated nuclear morphology. This suggests that either our TM1 cluster contains more than 1 TM cell subtype or that the morphology of a single TM1 cell subtype is influenced by local environment. This needs to be resolved by future studies. Compared to the other TM cell subtypes, TM1 cell pathways are enriched in both ECM structural molecules, such as collagens and integrins, and TGF-β signaling, which increases ECM production (*Munger and Sheppard, 2011*). Notably, excessive TGF-β signaling and ECM production can lead to fibrosis and elevate IOP (*Fleenor et al., 2006*; *Fuchshofer and Tamm, 2009*; *Meng et al., 2016*; *Frangogiannis, 2020*; *Biernacka et al., 2011*; *Fuchshofer and Tamm, 2012*). TGFβ2 has also been shown to cause ocular hypertension both in vivo (*McDowell et al., 2013*; *Shepard et al., 2010*; *Rudzitis et al., 2025*) and ex vivo (*Fleenor et al., 2006*; *Gottanka et al., 2004*) and is elevated in POAG patients (*Tripathi et al., 1994*; *Min et al., 2006*). These data suggest that TM1 cells play an important role in the constant remodeling of the ECM. However, dysregulation of these pathways can be pathogenic.

TM2 cells molecularly resemble previously sequenced human TM beam cells (*van Zyl et al., 2020*; *Patel et al., 2020*). This includes overlap of expression of top TM2 signature gene expression with sequenced human beam cells. Consistent with a beam cell identity, TM2 cells are located more towards the mid and inner TM than most TM1 cells. TM2 cells have an elongated nuclear morphology, consistent with the previously established beam cell morphology of cells in the regions they occupy (*Stamer and Clark, 2017*; *Smith et al., 2001*; *Overby et al., 2014*). Beam cells have endothelial-like properties, such as maintaining the patency of the AH outflow tract through the production of anti-thrombotic molecules. Proteases and molecules classically known as antithrombotic and thrombolytic maintain fluid flow through the TM by preventing build-up of protein aggregates to maintain fluid drainage (*Acott and Kelley, 2008*; *Stamer and Acott, 2012*). Such molecules include glycosamino-glycans, like heparin, the serine protease tPA, and matrix metalloprotease that are all highly expressed in TM2 cells. Beam cells are also phagocytic, and this activity, along with turnover of ECM, further cleans the TM (called filtering) to enhance fluid drainage (*Stamer and Clark, 2017*). Human beam cells also participate in antigen presentation and modulation of inflammation through cytokines and major histocompatibility proteins (*Stamer and Clark, 2017*; *Shifera et al., 2010*), consistent with TM2 cells being enriched for these immune pathways/molecules. Additionally, our data suggest that signaling from TM2 cells to SC endothelial cells is mediated by soluble molecules including *Vegfa*, *Edn3*, and *Angpt1*, which are critical for SC maintenance and function. This links TM2 cells to ocular development and glaucoma. VEGF receptor and ANGPT signaling genes modulate SC development/ function as well as contribute to developmental and primary open-angle glaucoma (*Kizhatil et al., 2014*; *Khawaja et al., 2018*; *MacGregor et al., 2018*; *Thomson et al., 2020*; *Souma et al., 2016*; *Kabra et al., 2017*; *Aspelund et al., 2014*).

TM3 cells also have a location and elongated nuclear morphology resembling beam cells with enriched localization in the inner, anterior TM. TM3 cell-enriched processes include actin binding and modulation of the actomyosin system, which are known to modulate outflow resistance and IOP (*Rao et al., 2001*; *Yu et al., 2008*). As such, TM3 cells highly express a large percentage of glaucoma-associated genes involved in actin binding. Importantly, TM3 cells are enriched for various mitochondrial metabolic and antioxidant pathways, including those associated with IOP elevation and glaucoma (*Aboobakar et al., 2023*; *Khawaja et al., 2016*). Inhibiting metabolic pathways alters AH outflow resistance (*Bermudez et al., 2017*). Since maintaining contractile tone is an energy-intensive process (*DeWane et al., 2021*), this could explain the enrichment of energy metabolism pathways in TM3 cells compared to other TM cells.

## TM3 cells have a unique susceptibility to mitochondrial dysfunction

In addition to being enriched in mitochondrial/ energy metabolism processes, TM3 cells express *Lmx1b* at significantly higher levels than both the other TM cell subtypes and other limbal cells. Importantly, in heterozygous mutant *V265D* mice, TM3 cells had pronounced gene expression changes that implicate mitochondrial dysfunction but that were absent or much lower in other cells including TM1 and TM2. In sections of the inner TM (enriched for TM3 cells), we show that the inner membrane folds known as cristae are disrupted or lacking in mutant mitochondria. Cristae disruptions are well estab-lished to disrupt cellular metabolism, impair oxidative phosphorylation, reduce ATP synthesis, alter mitochondrial membrane potential, and elevate ROS production (*Baker et al., 2019*; *Jenkins et al., 2024*; *Golombek et al., 2024*; *Huang et al., 2023*; *Ježek et al., 2023*). Together, these data suggest that mutation of *Lmx1b* has a primary effect on metabolism in TM3 cells that subsequently leads to IOP elevation and then glaucoma. Relevantly, homozygous conditional knockout of both *Lmx1b* and *Lmx1a*, as well as siRNA knockdown of *Lmx1b* alone, impacts mitochondrial functions in neurons and may contribute to Parkinson's disease (*Jiménez-Moreno et al., 2023*; *Doucet-Beaupré et al., 2016*; *Bergman et al., 2009*). Our data extend these published findings by showing that inheritance of a single dominant mutation in *Lmx1b* similarly affects mitochondria in TM cells. Although our studies show a clear effect of the *Lmx1b* mutation on TM mitochondria, it is not clear whether these are primary effects of the mutation or secondary effects. Future studies are needed to determine whether LMX1B directly modulates mitochondrial function in *V265D* mutant TM3 cells, which will require tech-nological advances to define the molecular etiology and nature of mitochondrial defects in *Lmx1b* mutant TM cells, their relationship to outflow dysfunction/IOP elevation, and the effects of NAM on TM cell mitochondria performance.

In addition to modulating mitochondria, LMX1B was recently identified as a TF that regulates responses to cell stress. This includes promotion of autophagy under stress conditions to enable recycling of cellular resources to maintain cellular functions and enable survival (*Jiménez-Moreno et al., 2023*). Thus, mutation of *Lmx1b* may also result in a deficiency of the beneficial autophagic response to stress (*Jiménez-Moreno et al., 2023*). Although further experiments are needed, dysfunctional autophagy may further prolong and exacerbate TM cell stress, resulting in more extensive depletion of cellular resources, exacerbated cellular dysfunction, and IOP elevation. Other consequences of the *Lmx1b* mutation that were evident in the *V265D* cells may also contribute to IOP elevation, including the overexpression of genes associated with ribosomes and calcium signaling, and depletion of genes involved with ECM synthesis and metabolism. However, these processes were not limited to TM3 cells or even to cell types that express detectable *Lmx1b*, suggesting that they are secondary damaging processes that are subsequent to the initiating, *Lmx1b*-induced perturbations in TM3 cells. Thus, we hypothesize that mitochondrial abnormalities in TM3 cells are of primary importance to IOP elevation in *V265D* glaucomatous mice.

## Nicotinamide protects against IOP elevation in *Lmx1b^V265D*-mutant mice

Based on our findings in TM3 cells, we tested if mitochondrial and metabolic disturbances are important in IOP elevation by administering nicotinamide (NAM). NAM is a form of vitamin B3 that is well-established to boost NAD$^+$ levels, promote mitochondrial health and energy metabolism, relieve oxidative stress and thus promote cellular resilience (*Williams et al., 2017b*; *Jang et al., 2012*; *Verdin, 2015*; *Mouchiroud et al., 2013*; *Gomes et al., 2013*; *Imai and Guarente, 2014*; *Song et al., 2021*; *Kang and Hwang, 2009*). Supporting our hypothesis of primary metabolic dysfunction in TM3 cells, NAM treatment strongly protected *V265D* mice from IOP elevation. Thus, treatments that support normal metabolism may protect from IOP elevation in individuals with *LMX1B* variants including both developmental glaucoma and POAG patients (*Gharahkhani et al., 2021*; *Khawaja et al., 2018*; *MacGregor et al., 2018*; *Choquet et al., 2018*; *Gao et al., 2018*; *Sweeney et al., 2003*; *Chen et al., 1998*; *Vollrath et al., 1998*; *Choquet et al., 2017*). As NAM and other NAD$^+$ boosting treatments are also known to be directly neuroprotective in glaucoma (*Williams et al., 2017b*; *Hui et al., 2020*; *Williams et al., 2017a*; *Williams et al., 2018*; *de Moraes et al., 2022*), our current data suggest that such treatments will have a double benefit for patients. As mitochondrial defects are becoming broadly associated with glaucoma risk and progression (*Abu-Amero et al., 2006*; *Lo Faro et al., 2021*; *Petriti et al., 2024*; *Venkatesan and Bernstein, 2025a*; *Venkatesan et al., 2025b*), these treatments could have wide-ranging potential to complement and augment existing IOP-lowering therapeutics through a completely distinct mechanism.

Our results provide a thorough molecular characterization of mouse TM cells, providing much-needed molecular information on subtype specializations in regulating IOP and the roles of specific subtypes in glaucoma. This comprehensive TM atlas provides a new foundation to guide development of new glaucoma treatments. We validate its utility by implicating metabolic changes in a single TM cell subtype in glaucoma, with success of a metabolic treatment in a glaucoma model.

# Materials and methods

## Key resources table

| Reagent type (species) or resource | Designation | Source or reference | Identifiers | Additional information |
| --- | --- | --- | --- | --- |
| Strain, strain background (*Mus musculus*) | C57BL/6J | The Jackson Laboratory | Strain #000664 RRID:IMSR_JAX:000664 | |
| Strain, strain background (*M. musculus*) | 129/Sj | PMID:33462143 | Stock #003884 | |
| Genetic reagent (*M. musculus*) | *Lmx1b^V265D* | PMID:24809698 | EM:00114 | Originally described as *Lmx1b^Icst* |
| Antibody | Rat monoclonal CD31 | eBioscience | 14-0311-82 RRID:AB_467201 | IF sections (1:200) |
| Antibody | Goat polyclonal VE-Cadherin | R&D Systems | AF-1002 RRID:AB_2077789 | IF sections (1:200) |

*Continued on next page*

*Continued*

| Reagent type (species) or resource | Designation | Source or reference | Identifiers | Additional information |
|---|---|---|---|---|
| Antibody | Rat monoclonal CD34 | eBioscience | 14-0341-81 RRID:AB_467209 | IF sections (1:100), TSA |
| Antibody | Rat monoclonal Chitinase 3-like 1 | R&D Systems | MAB2649-SP RRID:AB_2081263 | IF sections (1:100), TSA |
| Antibody | Rat monoclonal LY6C | Abcam | ab54223 RRID:AB_881384 | IF sections (1:100) |
| Antibody | Rabbit polyclonal Mu Crystallin | Proteintech | 12495–1-AP RRID:AB_2084620 | IF sections (1:100), TSA |
| Antibody | Goat polyclonal MYOC | R&D Systems | AF2537 | IF sections (1:200), IF Whole mount (1:50) |
| Antibody | Rabbit polyclonal TFAP2B | Novus Biologicals | NBP1-89063 | IF sections (1:200) |
| Antibody | Rabbit polyclonal αSMA | Abcam | ab5694 RRID:AB_2223021 | IF sections (1:200), IF Whole mount (1:50) |
| Antibody | Mouse monoclonal αSMA | Sigma Aldrich | A5228 RRID:AB_262054 | IF sections (1:500) |
| Commercial assay or kit | RNAscope | Advanced Cell Diagnostics | 323100 | |
| Commercial assay or kit | Tyramide signal amplification | Akoya Biosciences | NEL744001KT | |
| Chemical compound, drug | Niacinamide | Sigma-Aldrich | N5535 | Nicotinamide synonym |
| Software, algorithm | Imaris | Oxford Instruments | RRID:SCR_007370 | 9.5.1 |
| Software, algorithm | Viking | http://connectomes.utah.edu | RRID:SCR_005986 | |
| Software, algorithm | R | The R Project for Statistical Computing | RRID:SCR_001905 | 4.1.0 |
| Other | DAPI stain | Thermo Fisher Scientific | 62248 | (1:1000) |

## Animal husbandry and ethics statement

Experimental animals were either C57BL/6J (Stock# 664) or 129/Sj (unique substrain of 129) (*Tolman et al., 2021*) inbred mice. The *Lmx1b$^{V265D}$* mutation was discovered in an N-ethyl-N-nitrosourea (ENU) mutagenesis screen (*Cross et al., 2014*; *Tolman et al., 2021*; *Thaung et al., 2002*). These mice were then backcrossed to the C57BL/6J (Stock #000664) for at least 30 generations. All mice were treated in accordance with the Association for Research in Vision and Ophthalmology's statement on the use of animals in ophthalmic research. The Institutional Animal Care and Use Committee of either Columbia University (protocol numbers: AC-AABD0557 and AC-AABE9554) or Duke University (protocol number: A226-21-11) approved all experimental protocols performed at their respective institutions. Mice were maintained on PicoLab Rodent Diet 20 (5053, 4.5% fat) and provided with reverse osmosis-filtered water. Mutant and control littermates were housed together in cages containing ¼-inch corn cob bedding, covered with polyester filters. The animal facility was maintained at a constant temperature of 22°C with a 14 hr light and 10 hr dark cycle.

## Genotyping of the *Lmx1b* allele

*Lmx1b$^{V265D}$* and *Lmx1b$^+$* genotypes were determined by direct Sanger sequencing of a specific PCR product. Genomic DNA was PCR amplified with forward primer 5′–ACACAAGGCTCTGCCTCCT–3′ and reverse primer 5′–CATGACCCACTGCTATCACC–3′ using the following program: (1) 94°C for 3 min; (2) 94°C for 30 s; (3) 57°C for 30 s; (4) 72°C for 1 min; (5) repeat steps 2–4 35 times; and (6) 72°C for 5 min. PCR products were purified and sequenced by the Genewiz (Azenta Life Sciences).

## Slit-lamp examination

Anterior eye tissues were initially examined approximately at 2 months of age. Balanced groups of males and females were examined. Photographs were taken with a 40× objective lens. Phenotypic evaluation included the same disease features found in past studies of *Lmx1b*-mutant mice (*Cross et al., 2014*; *Tolman et al., 2021*). Anterior chamber deepening was graded based on a

semiquantitative scale of phenotype being not present, mild, moderate, or severe in presentation (*Tolman et al., 2021*). Experimenters were masked to mouse genotype and drug treatment. Groups were compared statistically by Fisher's exact test. n>30 eyes (biological replicates) examined in each group.

## IOP measurement

IOP was measured with the microneedle method as previously described in detail (*John et al., 1997*; *Savinova et al., 2001*). Prior to cannulation, mice were acclimatized to the procedure room and anesthetized via an intraperitoneal injection of a mixture of ketamine (99 mg/kg; Ketlar, Parke-Davis, Paramus, NJ, USA) and xylazine (9 mg/kg; Rompun, Phoenix Pharmaceutical, St Joseph, MO, USA) immediately prior to IOP assessment, a procedure that does not alter IOP in the experimental window (*Savinova et al., 2001*). IOP was measured at both 2 months and 3 months of age in WT and $Lmx1b^{V265D/+}$ eyes. Balanced groups of males and females were examined. During each IOP measurement period, eyes of independent WT B6 mice were assessed in parallel with experimental mice as a methodological control to ensure proper calibration and equipment function. *Lmx1b* mutant groups were compared by ANOVA followed by Tukey's honestly significant difference. n≥35 eyes examined in each WT group and n≥40 in *V265D* mutant groups. Cohort sizes for all *V265D* mutant groups were determined using power calculations based on previous IOP data, ($\alpha$=0.05, power = 0.8). Sample sizes required to reach that power and $\alpha$ levels for IOP analysis determined the cohort sizes as IOP analysis required the largest sample sizes.

## NAM administration

Nicotinamide (NAM or niacinamide; Sigma-Aldrich, St. Louis, Missouri) was dissolved in the standard institutional drinking water to a dose of 550 mg/kg/day (low dose) based on the average volume adult mice consume. Untreated groups received the same drinking water without NAM. Water was changed once per week. Treatment was started at postnatal day 2 and continued throughout the experiment. Births were checked daily between 9 a.m. and 12 p.m. to determine the pup's age.

## Sequencing

### Preparation of ocular cells

#### Columbia animals

Eyes were pooled from four animals to generate one sample (eight eyes total per pool). Two independent samples were processed. Mice were 2 months of age. Sex was balanced in each group.

#### Duke animals

One sample was generated from a pool of four animals (eight eyes total). All mice were male. Mice were 3 months of age.

#### Dissections

After euthanizing the animals, eyes were enucleated into fresh Dulbecco's Modified Eagle Medium (DMEM; Gibco) and subsequently dissected in DMEM. All dissection tools and surfaces were treated with RNAse zap (Ambion). For dissections, we removed the posterior eye by cutting between the limbus and sclera. The lens was then removed. Finally, we made a small circular cut in the anterior cornea, removing the middle one-third of the area. Only limbal tissues were collected, and all other tissues were removed. The remaining limbal tissue was minced and prepared for digestion.

## Single-cell RNA sequencing

### Single-cell RNA sequencing (Columbia University)

Tissues were digested enzymatically using Papain and Deoxyribonuclease I (Worthington Biochemical Corporation, Cat. no. LK003153) for 20 min at 37°C and stopped using Earl's balanced salt solution (EBSS, Thermo Fisher Scientific, Cat no. 24010–043). Cells were triturated using an 18-gauge needle, centrifuged at 300 × *g* at 4°C, washed with cold DMEM, and filtered using a 100 µm Flowmi Cell Strainer (H13680-0040, Bel-Art). Cells were resuspended in cold DMEM and placed on ice immediately. Cells were counted using the Countess II automated cell counter and processed for 10x library preparation immediately (see below).

### Single-cell RNA sequencing (Duke University)

Single cell dissociation was performed using Collagenase IV (Worthington Biochemical Cat. no. LS004188), Dispase II (Sigma Cat. No. D4693) and DNAse I (Roche Cat no. 50225500) for 60 min and then trypsin/EDTA for 10 min at 37°C. Cells were triturated using a 1 ml pipette tip, centrifuged at 300 × g at 4°C, washed with cold DMEM containing 10% FBS, and filtered using a 70-μm cell strainer (Thermo Fisher). Cells were resuspended in cold DMEM containing 5% FBS and placed on ice immediately and sent to Duke Genome Core Facility. Cells were counted using the Countess II automated cell counter (the cell viability was 88%) and then immediately processed (see below).

### Multiome (Columbia University)

Limbal strip samples for multiome analysis were snap-frozen on dry ice immediately following dissection and stored at –80°C until processing. To isolate nuclei, tissue was homogenized in 100 μL of chilled lysis buffer (10 mM Tris-HCl, pH 7.4, 10 mM NaCl, 3 mM MgCl2, 0.1% NP40, 0.1% Tween20, and 0.01% digitonin – from *Corces et al., 2017* supplemented with 1% BSA). Afterward, samples were incubated for 5 min on ice, gently agitated by pipetting, and incubated for an additional 10 min on ice. We then added 1 mL of chilled wash buffer (10 mM Tris-HCl, pH 7.4, 10 mM NaCl, 3 mM MgCl2, 0.1% Tween 20, 1% BSA) to the cells and centrifuged at 500 rcf for 5 min at 4°C. Cells were resuspended in chilled Diluted Nuclei Buffer (PN-2000153, 10x Genomics) and filtered through a 40 μm Flowmi Cell Strainer (H13680-0040, Bel-Art). Cells were resuspended in cold DMEM and placed on ice immediately. An aliquot of nuclei suspension was stained with SYTOX green to count nuclei using Countess II automated cell counter and processed for 10x library preparation immediately (see below).

Single-cell and single-nucleus RNA-seq was performed at the single-cell sequencing core in Columbia Genome Center (Columbia University) or Duke Molecular Genomics Core (MGC, Duke University). Single cells or nuclei were loaded into chromium microfluidic chips with v3 chemistry and barcoded with a 10x chromium controller (10x Genomics). RNA from the barcoded cells was reverse-transcribed, and sequencing libraries were constructed with a Chromium Single Cell v3 reagent kit (10x Genomics). Sequencing at both institutions was performed on NovaSeq 6000 (Illumina).

## Sequencing data processing

### Single cell and single nucleus sequencing

Raw reads were mapped to the mm10 reference genome by 10x Genomics Cell Ranger pipeline. 'Seurat' (version 4.0.1; *Hao et al., 2021*) was used to conduct all single-cell and single-nucleus sequencing analyses. Briefly, the dataset was filtered to contain cells with at least 200 expressed genes and genes with expression in more than three cells. Cells were also filtered for mitochondrial gene expression (<20% for single-cell and <5% for single-nucleus). The dataset was log-normalized and scaled. Biological incompatibility based on gene expression was used to identify doublets. For the scRNA-seq dataset generated at Columbia scRNA-seq, we performed unsupervised clustering followed by comparison to previous annotations (*Thomson et al., 2021*; *Balasubramanian et al., 2024*; *van Zyl et al., 2020*; *van Zyl et al., 2022*; *Patel et al., 2020*). A resolution parameter of 0.03 was used to match these data and previous annotations. Cluster 1 was then subclustered using unsupervised methods with a resolution parameter of 0.3. All cluster identities are based on a combination of known marker genes and validation experiments. For additional datasets (Duke scRNA-seq and Columbia snRNA-seq), parameters were manually adjusted to cluster cells based on the Columbia scRNA-seq dataset. Marker genes were identified using the Wilcoxon test implemented in Seurat using default parameters.

### Multiome

Single-nucleus multiome data were quantified by 10x CellRanger and preprocessed according to integratedSeurat v4 (*Hao et al., 2021*) and Signac (*Stuart et al., 2021*). We filtered cells which have <500 and>80,000 ATAC reads, have <500 and>25000 RNA reads, and have >10% mitochondrial reads. RNA-seq processing was performed using the same methods as scRNA-seq (above). ATAC-seq processing was performed using ArchR (version 1.0.2; *Granja et al., 2021*), where we used default latent semantic indexing (LSI) approach to derive low-dimensional embedding followed by single-cell

neighborhood construction and graph-based clustering using the top 50 LSI dimensions excluding the first dimension. We assigned cell types using a similar process based on marker peak accessibility and transferred cell type labels to the ATAC data.

### Lmx1b<sup>V265D</sup> analysis

Data from limbal cells with the *Lmx1b*<sup>V265D</sup> mutation (C57BL/6J background) were integrated with Wild-Type (WT) cells from the C57BL/6J strain only. Multiple approaches were used for normalizing data for integration in Seurat, including log normalization and SC transform. The relative number of cells in each cell cluster between WT and *V265D* mutant cells changed based on the integration technique. Therefore, we chose not to assess TM or other cell proportions between genotypes. Log normalization was selected for downstream analysis based on the log-transformed integrated WT clusters being more molecularly similar to the WT only clusters compared with the SC transform results.

## Statistical data analysis

### Hierarchical clustering

TM-containing cells were assessed by hierarchical clustering based on the similarity of the expression of the top 500 marker genes (ranked by P value) of each individual cluster. This analysis filtered out low-expressing genes (expressed in >10% of cells, with a logFC >0.25 compared to other cells).

### GO analysis

Gene ontology and pathway enrichment analysis were performed by comparing the differentially expressed genes in three scenarios:

1. All TM cells versus all other sequenced cells.
2. Each individual TM cell subtype against other TM cell subtypes.
3. WT versus *V265D* TM cells.

For comparisons of all TM cells to all sequenced cells, pathway enrichment was compared against a background expression universe of genes expressed in any sequenced cell. For comparisons of individual TM cell subtypes, only genes expressed in TM cells were used as a background (p value cut-off 0.01; *Timmons et al., 2015*).

Gene set enrichment analysis (GSEA) was performed using the same set of differentially expressed genes from the above comparisons with the GseGO function. All pathways were assigned an enrichment score (<0, underrepresented; >0, enriched). Pathways that were significantly enriched or underrepresented were also analyzed (P value cut-off 0.01). The R package ClusterProfiler was used for analysis (*Yu et al., 2012*).

### Module score

From genome-wide association studies of IOP elevation (*Khawaja et al., 2018*; *MacGregor et al., 2018*; *Gao et al., 2018*; *Choquet et al., 2017*; *van Koolwijk et al., 2012*; *Hysi et al., 2014*; *Springelkamp et al., 2017*; *Nag et al., 2014*; *Springelkamp et al., 2015*; *Ozel et al., 2014*; *The Blue Mountains Eye Study (BMES) and The Wellcome Trust Case Control Consortium 2 (WTCCC2), 2013*) or primary open-angle glaucoma (*Gharahkhani et al., 2021*; *Choquet et al., 2018*; *Shiga et al., 2018*; *Bonnemaijer et al., 2018*; *Verma et al., 2024*; *Bailey et al., 1996*; *Craig et al., 2020*), we curated a list of genes associated with the disease. Seurat's AddModuleScore function was applied to assess glaucoma disease risk for each cell type, using default parameters (24 bins for aggregation with 100 control features per analyzed feature).

### Predicted ligand-receptor interactions

Predicted ligand-target links between interacting cells were identified using LRLoop (*Xin et al., 2022*), which was developed based on NicheNet (*Browaeys et al., 2020*). Briefly, the expression of genes in cell types is linked to a database of signaling and gene regulatory networks curated from prior information, enabling viable predictions of potential interactions between cell types. These interactions were calculated using target molecules from one cell type and ligands from multiple cell types.

The top interactions, regardless of the cell type origin of the ligand, are represented in a Circos plot. Ligands were assigned to an individual TM cell subtype/subtypes based on the following criteria:

Exclusively assigned to a TM cell subtype if expressed >1 transcript per ten thousand in a given cell type and expressed on average 4X more in that subtype compared to all other TM cell subtypes.

Assigned to multiple TM cell subtypes if expression is >1 transcript per ten thousand in multiple cell types and average ligand expression is <4X between those TM cell subtypes.

If ligand expression is low (<1 transcript per ten thousand) across TM cell subtypes, the ligand is assigned to all subtypes that express the ligand.

## Comparison to published datasets

To compare TM cell clusters across datasets, the top 10 marker genes (by p value) for TM cell subtypes in our dataset were compared to published marker genes in other TM cell datasets (*Thomson et al., 2021*; *van Zyl et al., 2020*; *van Zyl et al., 2022*; *Patel et al., 2020*; *Ujiie et al., 2023*). Comparisons to both human and mouse TM cell data in the van Zyl et al. dataset were performed using the Broad Institute single-cell browser:

https://singlecell.broadinstitute.org/single_cell/study/SCP780.

## ATAC

To refine ATAC-seq analysis, we applied ArchR (*Granja et al., 2021*) to remove any additional doublet cells and iteratively estimated LSI embedding using 500 bp peaks tiled across the genome, followed by ArchR clustering and UMAP projection workflows to generate ATAC-seq cell clusters and two-dimensional embeddings (same cluster resolutions as above). Using marker gene expression, clusters were named and related to the single-cell cluster identities. Using single-cell RNA-seq data collected earlier with annotated cell types, we projected single-cell ATAC-seq data onto the reference using ArchR's addGeneIntegrationMatrix function with default parameters. The integrated data with RNA-seq defined clusters were used for peak calling with MACS2, followed by marker peak identification (getMarkerFeatures) and TF enrichment (addMotifAnnotations) using the CisBP database (*Weirauch et al., 2014*). Finally, TF activity was quantified using chromVAR (*Schep et al., 2017*), and gene-peak links were inferred using the addPeak2GeneLinks function with default parameters. All significant TFs (VAR >1) for each TM cell subtype were filtered by average RNA expression within the entire cluster using the Columbia University scRNA-seq data (better read depth than snRNA-seq). All significant TFs with no RNA expression were removed and all TFs with expression >0.5 counts per 10,000 were prioritized for analysis. For a schematic of the analysis workflow, see *Figure 4A*. We then performed integrative analysis correlating the activity of each TF with its gene expression profile (both directly measured from multiome data and integrated expression data from reference scRNA-sequencing data). We used ArchR's correlateMatrices function, which reports Pearson correlation of normalized counts (TF activity matrix and gene expression / gene integration matrix) across 100 randomly subsampled neighborhoods of cells. We designated influential TFs as those with correlation >0.5, adjusted p-value of correlation <0.01, and chromVAR deviation above 75% of all TFs.

### ATAC tracts

We chose marker genes (TM1: *Fn1*, TM2: *Gpc3*, TM3: *Acta2*) and plotted accessibility tracks. Motif footprinting analysis was performed using ArchR getFootprints function using CisBP-defined motif positions for RNA-defined cell types with default parameters.

## Immunofluorescence and in situ hybridization

### Sections with antibody staining

Enucleated adult eyes were fixed for 1 hr at 4°C in 4% paraformaldehyde (PFA, Electron Microscopy Science, Hatfield, PA) prepared in phosphate-buffered saline (1X PBS, 137 mM NaCl, 10 mM phosphate, 2.7 mM KCl, pH 7.4). Following fixation, a small window was made through the optic nerve head so that eyes would not shrivel during dehydration. Eyes were dehydrated in a 30% sucrose (Sigma-Aldrich, St. Louis, Missouri) in 1X PBS at 4°C until eyes sunk to the bottom of a 2 ml glass vial. Once sunken, eyes were embedded in optimal cutting temperature embedding medium (Thermo

Fisher Scientific, Waltham, MA) and flash frozen on dry ice. Eyes were cryosectioned at 10 µm thickness in the sagittal plane. Sections were evenly spaced throughout the peripheral and central ocular regions.

Sections were washed three times with 1× PBS with 0.3% Triton X-100 for 5 min to quench residual fixation. The sections were then incubated with 10% Donkey serum (Sigma-Aldrich, St. Louis, Missouri) and 0.3% Triton X-100 in 1× PBS (blocking buffer) at room temperature for 1 hr to block nonspecific binding of antibody and to permeabilize the tissue. The sections were then incubated with primary antibodies (Key resources table) in 200 µl blocking buffer overnight at 4°C. The sections were then washed three times for 5 m with 1× PBS with 0.3% Triton X-100. Primary antibodies were detected with the appropriate species-specific secondary antibody (all Alexa 488, 594, or 647 at 1:1000 dilution, Life Technologies, Grand Island, NY) diluted in 1× PBS with 0.3% Triton X-100 for 2 hr at room temperature. The secondary detection solution also had DAPI at 1:1000 (Thermo Fisher Scientific, Waltham, MA) to label nuclei. The immunostained sections were washed three times for 5 m in 1× PBS with 0.3% Triton X-100. Sections were then cover-slipped using Fluoromount (Sigma, St. Louis, MO).

## Tyramide signal amplification (TSA)

We performed a TSA reaction for a subset of antibodies (Key resources table). TSA was performed on slides containing serial transverse cryosections of enucleated mouse eyes using the Akoya Biosciences TSA Plus Cyanine 3 detection kit (NEL744001KT). Freshly cut sections were allowed to come to room temperature and were then quenched for 10 min in a 1% hydrogen peroxide, 10% methanol in 0.01 M phosphate buffered saline solution (1X PBS). The slides were rinsed in 1X PBS and allowed to block for 1 hr at room temperature in a blocking solution of 10% donkey serum, 0.3% Triton-X 100 in 1X PBS. Primaries for TSA were diluted 1:100 in blocking solution and approximately 200 µl were added to the slides overnight at 4 °C. The slides were then rinsed using a 0.3% Triton-X 100 in 1X PBS solution before being blocked in Tris-NaCl-blocking (TNB) buffer for 1 hr at room temperature. Approximately 200 µl of Horseradish peroxidase-conjugated secondaries diluted 1:200 in TNB buffer were applied to the slides for 2 hr at room temperature. The slides were subsequently rinsed in Tris-NaCl-Tween (TNT) buffer. TSA Plus Working Solution was made by diluting the TSA Plus Stock Solution 1:50 in 1X Amplification Diluent. The TSA Plus Working Solution was added at a volume of 200 µl for 7 min at room temperature. The slides were then washed in TNT buffer, and approximately 200 µl of non-amplified primaries for counterstaining diluted 1:200 in 10% donkey serum, 0.3% Triton-X 100 in 1X PBS blocking solution was applied overnight at 4 °C. Secondary antibody detection and mounting were performed as above.

## in situ hybridization

We used the RNAscope Multiplex Fluorescent Reagent Kit v2-Mm (Advanced Cell Diagnostics a Bio Techne Brand, Newark, CA). Probes for mouse *Lypd1* (#318361-C2), *Inmt* (#486371-C2), *Edn3* (#505841-C1), *Acta2* (α-SMA) (#319531-C3), and *Myoc* (#460981-C1) (also purchased from ACD Bio) were used. Fluorescent dyes Opal 690 (FP1497001KT) and Opal 620 (FP1495001KT) (Akoya Biosciences, Marlborough, MA) were used with each of these probes. Briefly, eyes from 3-month-old C57BL/6J were enucleated and fixed in 4% PFA for 24 hr at 4°C. A window was made to the back of the eye cup, lens removed, and the anterior cup placed in 30% sucrose overnight at 4°C. Tissue was cryopreserved in OCT and placed at –80°C. Sections were cut at 12 µm thickness using SuperFrost Plus slides (Fisher Scientific, Hampton, NH). RNAscope was performed over 2 days as per manufacturer's protocol.

## Whole mounts

Enucleated eyes were fixed for 2 hr at 4°C in 4% paraformaldehyde (PFA, Electron Microscopy Science, Hatfield, PA) prepared in phosphate-buffered saline (1× PBS, 137 mM NaCl, 10 mM phosphate, 2.7 mM KCl, pH 7.4). The postnatal eyes were dissected out following fixation. The anterior part of the eye was cut just posterior to the limbus (transition zone between cornea and sclera) as described in *Kizhatil et al., 2014*. Briefly, the iris, lens, ciliary body, and thin strip of retina were carefully removed to obtain the anterior eye cup. The anterior cup includes the cornea, limbus, and a small

portion of retinal pigmented epithelium. Four centripetal cuts were made to relax the eye cup and facilitate eventual mounting onto a slide after immunofluorescence.

Anterior eye cups were rinsed multiple times with 1× PBS. The eye cups were incubated with 3% bovine serum albumin and 1% Triton X-100 in 1× PBS (blocking buffer) at room temperature for 1 hr in 2 ml glass vials to block nonspecific binding of antibody and to permeabilize the tissue. The anterior cups were then incubated with primary antibodies (Key resources table) in 200 µl blocking buffer for 2 days, with rocking, at 4°C. The anterior cups were then washed three times over a 3 hr period with 1× PBS. The primary antibodies were detected with the appropriate species-specific secondary antibody (all Alexa 488, 594, or 647 at 1:1000 dilution, Life Technologies, Grand Island, NY) diluted in blocking buffer, which also had DAPI to label nuclei. The immunostained eye cups were washed three times over a 3 hr period in 1× PBS. Eye cups were then whole-mounted on slides in Fluoromount (Sigma, St. Louis, MO).

## Microscopy of drainage structures
### Sections
Microscopy of sectioned postnatal eye slides was performed using an LSM SP8 confocal microscope (Leica) and 40×1.1 NA water immersion objective.

### In situ
Images were taken using a Nikon Eclipse 90i confocal laser-scanning microscope (Melville). Z-stack images of 1 µm thickness, using the 40X objective lens, were taken of each section and converted into maximum projection images that were used for analysis.

### Whole mounts
Microscopy was performed using an LSM SP8 confocal microscope (Leica) using a 63×1.4 NA glycerol immersion objective. The Mark and Find mode was used to automate collection of images, generating a folder full of Z stacks (Z=0.3 µm) at various individual overlapping positions around the limbus. We examined n=6 eyes total. We imaged a certain distance around the ocular circumference, with an equal representation from each ocular quadrant (1.5–2.5 mm total).

## Postprocessing of whole-mount images
### 3D rendering
We followed the guidelines detailed in Kizhatil et al. for visualizing SC and TM in 3D. Individual confocal Z stacks (.lsm files) were processed directly using Imaris 9.5 (Oxford Instruments, Carteret, NJ). The maximum intensity projection setting in Surpass mode of Imaris was used for 3D rendering. Controls and experimental image sets were treated identically. Images were oriented so that structures of interest were visible. For whole-mounted eyes, multiposition Z stacks generated using the Mark and Find setting on the confocal microscope were first imported into Imaris and converted into Imaris files (.ims files). The resulting Imaris files were then stitched to generate a comprehensive Z stack encompassing the limbus using ImarisStitcher 9.5 (BitPlane AG, Zurich, Switzerland). The stitch was exported as an Imaris file and processed appropriately. For sections, images were taken in the Z-plane in focus. When making figures, the snapshot feature of Imaris was used to collect images at high resolution (1024×1024 pixels, 300 dpi).

## TM marker location analysis
We defined the TM as the region between the inner wall of Schlemm's canal (SC, outer TM) and the anterior chamber (inner TM), extending from the anterior edge of the pars plana (posterior TM) to the posterior edge of Descemet's membrane of the corneal endothelium (anterior TM). The SC was defined using a panel of validated markers that varied to allow for different antibody species and different secondary antibodies to be used in conjunction with the SC markers (Key resources table; *Kizhatil et al., 2014*). Sections were only included in the scoring analysis if they met the following criteria:

1. Absence of obvious sectioning artifacts or abnormalities in the SC or TM region.
2. Distinguishability of all landmarks defining the TM.

3. Acceptably low background signal and absence of staining for non-specific or non-biological entities within the section.

For each section, the TM was segmented and analyzed in two separate ways. The anterior-posterior distance was measured, and the TM was divided in half along this axis at the midway point (*Figure 2—figure supplement 2C–D*). Using the surface feature in Imaris, the area of individual marker staining was measured, and the total area in both the anterior and posterior zones was calculated. Because markers are expressed at different levels, we normalized across markers by calculating the percent of individual marker expression in the anterior and posterior halves within each section. These percentage values in anterior and posterior halves were compared both within markers and across all markers of a given TM cell subtype by Student's t-test. The same analysis was repeated for the inner-outer halves by measuring the inner-outer TM distance at the midpoint of the TM (anterior-posterior axis) and dividing the TM in half at this value. A total of 130–160 sections were examined for markers of each individual TM cell subtype. These sections were from 10 to 12 individual eyes per cluster. All eyes were from WT mice of C57BL/6J background and between 3 and 6 months of age.

Additional analysis was performed by selecting a subset of examined sections with staining for select markers (TM1: MYOC antibody, TM2: CRYM antibody, Inmt in situ, TM3: α-SMA antibody). These sections were of exceptionally high quality and had: (1) Anterior-posterior TM length between 125 and 200 µm. (2) Inner-outer TM distance is between 15 and 30 µm.

With these selected sections, we subdivided the TM into eight different zones along both inner/outer and anterior/posterior axes (*Figure 2—figure supplement 2F*). For each individual section, we used Imaris to measure the anterior-posterior distance and divide the TM into three equal sectors (anterior, central, and posterior). Within each sector, the average depth of the TM was measured in Imaris. This calculation was used to divide the central and posterior sectors into three zones each (inner, intermediate, and outer). Because the anterior TM depth is thinner, we divided the anterior sector into two zones (inner and outer). In total, there were eight distinct zones.

The percent of total marker area was calculated for each of the eight zones for each individual section as described above. Percent of marker area in each zone was compared as shown in *Figure 2* by one-way ANOVA followed by Tukey's honestly significant difference test.

## Scleral cell marker analysis

For CD34 and LY6C1 expression, we examined greater than 60 sections from 4 eyes (CD34) and 3 eyes (LY6C1) respectively. The observed locations of these markers within the iridocorneal angle are described.

## Whole mount marker location analysis

To analyze the expression of markers within the TM, we segmented the TM based on anatomy. Because various anatomical features, including the pars plana, were removed during the whole mount process, we used the SC (marked by PECAM) as the major anterior-posterior landmark. The TM was defined as the group of cells inner to the SC and not outside the anterior or posterior boundary of the SC.

## Nuclear morphology

Anterior segment whole-mounts were used to examine nuclear morphology in 3D (TM1: MYOC, TM3: α-SMA, all antibodies). The TM was segmented as described above and the 'Surface' feature in Imaris was used to demarcate TM cell nuclei. Nuclei that were analyzed were immediately adjacent to the immunolabeled TM marker and had to be well spaced from other nuclei. To ensure accuracy, we used manual inspection to exclude: (1) Nuclei that were very close to other nuclei or groups of nuclei and (2) nuclei of cells that were labeled with a Schlemm's canal endothelial cell marker (PECAM) and/or that projected into the lumen of SC. The 'Sphericity' tool was used to determine the degree of sphericity in Imaris. For each cell subtype, we assessed a total of 60 nuclei from 6 eyes, assessing the first 10 randomly selected nuclei that met our inclusion criteria in each eye. Groups were compared by Student's t-test.

## Electron microscopy

Mice were euthanized, and eyes were enucleated into DBG buffer containing 1X Dulbecco's phosphate-buffered saline with 5.5 mM D-glucose (Sigma-Aldrich). Eyes were immediately dissected

in DBG buffer to generate anterior segment eye cups, with centripetal relaxing cuts in the cornea to allow flattening (*Figure 1—figure supplement 1*). The tissue was fixed at room temperature for 30 min in a solution of 0.8% paraformaldehyde, 1.2% glutaraldehyde, and 0.08 M phosphate buffer (Smith-Rudt). Following fixation, each anterior segment was mounted on a slide and flattened under a coverslip overnight at 4°C. A 1 g weight was placed on top of the coverslip to aid in flattening the limbal region. The eyes were then washed three times in 1X PBS for 15 min each and stored in PBS under the same slide/coverslip setup to maintain flattening until embedding. Before further processing, individual limbal strips of the four quadrants of each were dissected from the surrounding cornea and sclera (*Figure 1—figure supplement 1*). These strips were post-fixed in 2% aqueous osmium tetroxide, rinsed in PBS, and dehydrated for epoxy resin embedding using EMBED 812 (Electron Microscopy Sciences). Embedding was performed while preserving the orientation of the limbal strip, with the TM (inner side) and cornea / sclera (outer side) clearly maintained. Embedded strips were imaged using computed tomography to locate the tissue, confirm TM orientation and flattening for *en face* sectioning, and to guide the sectioning process. Ultrathin, 70 nm, *enface* sections of the inner side of the limbal strip were taken using a Leica ultramicrotome (inner TM is the most superficial tissue being sectioned). Sections were placed onto formvar-coated slot grids then stained en grid using a combination of 5% Uranyl Acetate and Reynold's lead citrate (*Reynolds, 1963*) for 17 and 7 min, respectively. Large mosaic regions of the TM were imaged at ×5000 magnification (2.18 nm/px) using a JEOL 1400FLASH TEM operating at 80 kV, equipped with a Gatan OneView camera. Mosaics were then computationally reconstructed using custom nornir-buildscripts (https://nornir.github.io/) and initially visualized in the Software suite Viking (http://connectomes.utah.edu).

From each TM section, 20 cells with sampled nuclei were randomly selected from evenly spaced regions throughout the imaged area (approximately 25,000 µm² per eye). In the perinuclear area of each selected cell, up to 3 (typically 2–3) mitochondria were randomly selected and analyzed. The perimeter of the mitochondrial outer membrane (area) and inner membrane (cristae) was traced in ImageJ according to established protocols (*Lam et al., 2021*; *Neikirk et al., 2023*). The percentage of cristae area compared to the whole mitochondrial area was calculated (cristae fraction) for each mitochondrion. The number of mitochondria was counted in the perinuclear region of each analyzed cell. This number was normalized to an area of 100 µm² and averaged across cells. As each en face section sampled a large area of the TM, three sections from different eyes were examined for each genotype. All mice were 14 days old. A total of 178 mitochondria from WT eyes and 161 from *V265D* eyes were analyzed. Groups were compared by Student's t-test.

## Acknowledgements

The authors thank the members of the Simon John Laboratory for experimental and technical assistance, Drs. Krish Kizhatil and Chi Zhang for reading and editing the manuscript, the Institute of Comparative Medicine and Devanshi Ragha at Columbia University for animal and veterinary care, the JP Sulzberger Columbia Genome Center and Duke University Molecular Genomics Core for assistance with genomics studies, and the P30EY019007 Columbia Shared Resources grant for histological experiments.

## Additional information

### Funding

| Funder | Grant reference number | Author |
| --- | --- | --- |
| National Eye Institute | EY032507 | Simon WM John |
| National Eye Institute | EY011721 | Simon WM John |
| National Eye Institute | EY032062 | Simon WM John |
| National Eye Institute | EY018606 | Simon WM John |
| BrightFocus Foundation | CG2020004 | Simon WM John |

| Funder | Grant reference number | Author |
|---|---|---|
| BrightFocus Foundation | G2021007 | Revathi Balasubramanian |
| Columbia University | Precision Medicine Initiative | Simon WM John |
| New York Fund for Innovation in Research and Scientific Talent | NYFIRST; EMPIRE CU19-2660 | Simon WM John |
| National Eye Institute | EY028608 | W Daniel Stamer |
| National Eye Institute | EY022359 | W Daniel Stamer |
| National Eye Institute | P30EY019007 | Simon WM John |
| National Eye Institute | P30EY005722 | W Daniel Stamer |
| National Eye Institute | EY028927 | Bryan Jones |
| Research to Prevent Blindness | Career Development Award | Rebecca Pfeiffer |
| Knights Templar Eye Foundation | Career Starter Grant | Rebecca Pfeiffer |
| U.S. National Science Foundation | 2014862 | Bryan Jones |
| Research to Prevent Blindness | Unrestricted Research Grant | Bryan Jones W Daniel Stamer Simon WM John |
| Gabe Newell | Unrestricted Research Grant | Bryan Jones |
| Glaucoma Foundation | Award | Simon WM John |

The funders had no role in study design, data collection and interpretation, or the decision to submit the work for publication.

## Author contributions

Nicholas Tolman, Conceptualization, Data curation, Software, Formal analysis, Validation, Visualization, Writing – original draft; Taibo Li, Revathi Balasubramanian, Conceptualization, Data curation, Software, Formal analysis, Validation, Visualization, Writing – review and editing; Guorong Li, Data curation, Formal analysis, Investigation; Rebecca Pfeiffer, Data curation, Investigation, Project administration; Violet Bupp-Chickering, Data curation, Formal analysis, Validation, Investigation; Ruth A Kelly, Formal analysis, Validation, Investigation; Marina Simón, Formal analysis, Writing – review and editing; John Peregrin, Investigation; Christa Montgomery, Data curation, Project administration, Writing – review and editing; Bryan Jones, Supervision, Funding acquisition, Project administration; W Daniel Stamer, Jiang Qian, Conceptualization, Formal analysis, Supervision, Funding acquisition, Project administration, Writing – review and editing; Simon WM John, Conceptualization, Formal analysis, Supervision, Funding acquisition, Writing – original draft, Project administration, Writing – review and editing

## Author ORCIDs

Nicholas Tolman ⓘ https://orcid.org/0000-0002-3060-2999
Taibo Li ⓘ https://orcid.org/0000-0002-6624-9293
Revathi Balasubramanian ⓘ https://orcid.org/0000-0002-2209-0815
Rebecca Pfeiffer ⓘ https://orcid.org/0000-0002-4123-516X
Marina Simón ⓘ https://orcid.org/0000-0001-8133-8668
Christa Montgomery ⓘ https://orcid.org/0009-0004-2430-2772
Bryan Jones ⓘ https://orcid.org/0000-0001-5527-6643
W Daniel Stamer ⓘ https://orcid.org/0000-0002-2504-8997
Simon WM John ⓘ https://orcid.org/0000-0002-4319-0356

### Ethics

All mice were treated in accordance with the Association for Research in Vision and Ophthalmology's statement on the use of animals in ophthalmic research. The Institutional Animal Care and Use Committee of either Columbia University (protocol numbers: AC-AABD0557 and AC-AABE9554) or Duke University (protocol number: A226-21-11) approved all experimental protocols performed at their respective institutions.

Reviewer #1 (Public review): https://doi.org/10.7554/eLife.107161.3.sa1
Reviewer #3 (Public review): https://doi.org/10.7554/eLife.107161.3.sa2
Author response https://doi.org/10.7554/eLife.107161.3.sa3

## Additional files

### Supplementary files

MDAR checklist

Supplementary file 1. Insitutional comparison of TM cell pathway analysis.

Supplementary file 2. TM cell differentially expressed genes.

Supplementary file 3. All TM cell pathway analysis.

Supplementary file 4. Ligand-target interaction data.

Supplementary file 5. Glaucoma GWAS gene pathway enrichment.

Supplementary file 6. V265D mutant DEG TM cell pathway analysis.

Supplementary file 7. IOP data statistics.

### Data availability

Sequencing data have been deposited in GEO under the accession code GSE309500. Sequencing data from strain 129 mice was previously deposited in GEO under the accession code GSE272434. Sequencing data can be visualized in the Broad Institute's Single Cell Portal under accession number SCP3243. The code used in this study is available on Github RNA sequencing: https://github.com/nicktolman26/Trabecular-meshwork-single-cell-RNA-seq-analysis, (copy archived at *Tolman, 2026*), ATAC sequencing: https://github.com/taiboli/Adult-TM-ATAC, (copy archived at *Li, 2026*).

The following datasets were generated:

| Author(s) | Year | Dataset title | Dataset URL | Database and Identifier |
|---|---|---|---|---|
| Tolman N, Balasubramanian R, Li T, Li G | 2025 | Single-cell profiling of trabecular meshwork identifies mitochondrial dysfunction in a glaucoma model that is protected by vitamin B3 treatment | https://www.ncbi.nlm.nih.gov/geo/query/acc.cgi?acc=GSE309500 | NCBI Gene Expression Omnibus, GSE309500 |
| Tolman N, Li T, Balasubramanian R, Li G, Bupp-Chickering V, Kelly RA, Simón M, Peregrin J, Montgomery C, Stamer DW, Qian J, John SMW | 2025 | Single-cell profiling of trabecular meshwork identifies mitochondrial dysfunction in a glaucoma model that is protected by vitamin B3 treatment | https://singlecell.broadinstitute.org/single_cell/study/SCP3243/single-cell-profiling-of-trabecular-meshwork-identifies-mitochondrial-dysfunction-in-a-glaucoma-model-that-is-protected-by-vitamin-b3-treatment | Single Cell Portal, SCP3243 |

The following previously published dataset was used:

| Author(s) | Year | Dataset title | Dataset URL | Database and Identifier |
|---|---|---|---|---|
| Balasubramanian R, Kizhatil K, Li T, et al | 2024 | Transcriptomic profiling of Schlemm's canal cells reveals a lymphatic-biased identity and three major cell states | https://www.ncbi.nlm.nih.gov/geo/query/acc.cgi?acc=GSE272434 | NCBI Gene Expression Omnibus, GSE272434 |

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
