## [Editor Report · eLife Assessment]

This study provides a **fundamental** advancement in our understanding of trabecular meshwork cell diversity and its role in eye pressure regulation and glaucoma using multimodal single-cell analysis, spatial validation, and functional testing that go beyond the current state-of-the-art. The study demonstrates that mitochondrial dysfunction, specifically in one of three distinct cell subtypes (TM3), contributes to elevated IOP in a genetic mouse model of glaucoma carrying a mutation in the transcription factor Lmx1b. While the identification of TM3 cells as metabolically specialized is **compelling**, there is somewhat limited evidence directly linking mitochondrial dysfunction to the Lmx1b mutation in TM3 cells.

---

## [Referee Report · Reviewer #1 (Public review)]

Summary:

This study provides a comprehensive single-cell and multiomic characterization of trabecular meshwork (TM) cells in the mouse eye, a structure critical to intraocular pressure (IOP) regulation and glaucoma pathogenesis. Using scRNA-seq, snATAC-seq, immunofluorescence, and in situ hybridization, the authors identify three transcriptionally and spatially distinct TM cell subtypes. The study further demonstrates that mitochondrial dysfunction specifically in one subtype (TM3) contributes to elevated IOP in a genetic mouse model of glaucoma carrying a mutation in the transcription factor Lmx1b. Importantly, treatment with nicotinamide (vitamin B3), known to support mitochondrial health, prevents IOP elevation in this model. The authors also link their findings to human datasets, suggesting the existence of analogous TM3-like cells with potential relevance to human glaucoma.

Strengths:

The study is methodologically rigorous, integrating single-cell transcriptomic and chromatin accessibility profiling with spatial validation and in vivo functional testing. The identification of TM subtypes is consistent across mouse strains and institutions, providing robust evidence of conserved TM cell heterogeneity. The use of a glaucoma model to show subtype-specific vulnerability-combined with a therapeutic intervention-gives the study strong mechanistic and translational significance. The inclusion of chromatin accessibility data adds further depth by implicating active transcription factors such as LMX1B, a gene known to be associated with glaucoma risk. The integration with human single-cell datasets enhances the potential relevance of the findings to human disease.

Weaknesses:

Although the LMX1B transcription factor is implicated as a key regulator in TM3 cells, its role in directly controlling mitochondrial gene expression is not fully explored. Additional analysis of motif accessibility or binding enrichment near relevant target genes could substantiate this mechanistic link. The therapeutic effect of vitamin B3 is clearly demonstrated phenotypically, but the underlying cellular and molecular mechanisms remain somewhat underdeveloped-for instance, changes in mitochondrial function, oxidative stress markers, or NAD+ levels are not directly measured. While the human relevance of TM3 cells is suggested through marker overlap, more quantitative approaches, such as cell identity mapping or gene signature scoring in human datasets, would strengthen the translational connection.

Overall, this is a compelling and carefully executed study that offers significant advances in our understanding of TM cell biology and its role in glaucoma. The integration of multimodal data, disease modeling, and therapeutic testing represents a valuable contribution to the field. With additional mechanistic depth, the study has the potential to become a foundational resource for future research into IOP regulation and glaucoma treatment.

---

## [Referee Report · Reviewer #3 (Public review)]

Summary:

In this study, the authors perform multimodal single-cell transcriptomic and epigenomic profiling of 9,394 mouse TM cells, identifying three transcriptionally distinct TM subtypes with validated molecular signatures. TM1 cells are enriched for extracellular matrix genes, TM2 for secreted ligands supporting Schlemm's canal, and TM3 for contractile and mitochondrial/metabolic functions. The transcription factor LMX1B, previously linked to glaucoma, shows the highest expression in TM3 cells and appears to regulate mitochondrial pathways. In Lmx1bV265D mutant mice, TM3 cells exhibit transcriptional signs of mitochondrial dysfunction associated with elevated IOP. Notably, vitamin B3 treatment significantly mitigates IOP elevation, suggesting a potential therapeutic avenue.

This is an excellent and collaborative study involving investigators from two institutions, offering the most detailed single-cell transcriptomic and epigenetic profiling of the mouse limbal tissues-including both TM and Schlemm's canal (SC), from wild-type and Lmx1bV265D mutant mice. The study defines three TM subtypes and characterizes their distinct molecular signatures, associated pathways, and transcriptional regulators. The authors also compare their dataset with previously published murine and human studies, including those by Van Zyl et al., providing valuable cross-species insights.

Strengths:

(1) Comprehensive dataset with high single-cell resolution

(2) Use of multiple bioinformatic and cross-comparative approaches

(3) Integration of 3D imaging of TM and SC for anatomical context

(4) Convincing identification and validation of three TM subtypes using molecular markers.

Weaknesses:

(1) Insufficient evidence linking mitochondrial dysfunction to TM3 cells in Lmx1bV265D mice: While the identification of TM3 cells as metabolically specialized and Lmx1b-enriched is compelling, the proposed link between Lmx1b mutation and mitochondrial dysfunction remains underdeveloped. It is unclear whether mitochondrial defects are a primary consequence of Lmx1b-mediated transcriptional dysregulation or a secondary response to elevated IOP. Although authors have responded to this, the manuscript is not sufficiently altered to address these points. I would like to suggest that authors tone down mitochondrial connection with Lmx1b from the title and abstract, and clearly discuss that these events are associated, and future work is needed to dissect the role of mitochondria in this pathway.

Furthermore, the protective effects of nicotinamide (NAM) are interpreted as evidence of mitochondrial involvement, but no direct mitochondrial measurements (e.g., immunostaining, electron microscopy, OCR assays) are provided. It is essential to validate mitochondrial dysfunction in TM3 cells using in vivo functional assays to support the central conclusion of the paper. Without this, the claim that mitochondrial dysfunction drives IOP elevation in Lmx1bV265D mice remains speculative. Alternatively, authors should consider revising their claims that mitochondrial dysfunction in these mice is a central driver of TM dysfunction.

(2) Mechanism of NAM-mediated protection is unclear: The manuscript states that NAM treatment prevents IOP elevation in Lmx1bV265D mice via metabolic support, yet no data are shown to confirm that NAM specifically rescues mitochondrial function. Do NAM-treated TM3 cells show improved mitochondrial integrity? Are reactive oxygen species (ROS) reduced? Does NAM also protect RGCs from glaucomatous damage? Addressing these points would clarify whether the therapeutic effects of NAM are indeed mitochondrial.

---

## [Author Response]

The following is the authors’ response to the original reviews.

**Reviewer #1 (Public review):**
Summary:This study provides a comprehensive single-cell and multiomic characterization of trabecular meshwork (TM) cells in the mouse eye, a structure critical to intraocular pressure (IOP) regulation and glaucoma pathogenesis. Using scRNA-seq, snATAC-seq, immunofluorescence, and in situ hybridization, the authors identify three transcriptionally and spatially distinct TM cell subtypes. The study further demonstrates that mitochondrial dysfunction, specifically in one subtype (TM3), contributes to elevated IOP in a genetic mouse model of glaucoma carrying a mutation in the transcription factor Lmx1b. Importantly, treatment with nicotinamide (vitamin B3), known to support mitochondrial health, prevents IOP elevation in this model. The authors also link their findings to human datasets, suggesting the existence of analogous TM3-like cells with potential relevance to human glaucoma.Strengths:The study is methodologically rigorous, integrating single-cell transcriptomic and chromatin accessibility profiling with spatial validation and in vivo functional testing. The identification of TM subtypes is consistent across mouse strains and institutions, providing robust evidence of conserved TM cell heterogeneity. The use of a glaucoma model to show subtype-specific vulnerability, combined with a therapeutic intervention-gives the study strong mechanistic and translational significance. The inclusion of chromatin accessibility data adds further depth by implicating active transcription factors such as LMX1B, a gene known to be associated with glaucoma risk. The integration with human single-cell datasets enhances the potential relevance of the findings to human disease.

We thank the reviewers for their thorough reading of our manuscript and helpful comments.

Weaknesses:(1) Although the LMX1B transcription factor is implicated as a key regulator in TM3 cells, its role in directly controlling mitochondrial gene expression is not fully explored. Additional analysis of motif accessibility or binding enrichment near relevant target genes could substantiate this mechanistic link.

We show that the Lmx1b mutation induces mitochondrial dysfunction with mitochondrial gene expression changes but agree with the referee in that we do not show direct regulation of mitochondrial genes by LMX1B. Emerging data suggest that LMX1B regulates the expression of mitochondrial genes in other cell types [1, 2] making the direct link reasonable. Future work that is beyond the scope of the current paper will focus on sequencing cells at earlier timepoints to help distinguish gene expression changes associated with the V265D mutation from those secondary to ongoing disease and elevated IOP. Additional studies, including ATAC seq at more ages, ChIP-seq and/or Cut and Run/Tag (in TM cells) will be necessary to directly investigate LMX1B target genes.

As we studied adult mice, mitochondrial gene expression changes could be secondary to other disease induced stresses. Because we did not intend to say we have shown a direct link, we have now added a sentence to the discussion ensure clarity.

Lines 932-934: “Although our studies show a clear effect of the Lmx1b mutation on mitochondria, future studies are needed to determine if LMX1B directly modulates mitochondrial genes in V265D mutant TM cells”

(2) The therapeutic effect of vitamin B3 is clearly demonstrated phenotypically, but the underlying cellular and molecular mechanisms remain somewhat underdeveloped - for instance, changes in mitochondrial function, oxidative stress markers, or NAD+ levels are not directly measured.

We agree that further experiments towards a fuller mechanistic understanding of vitamin B3’s therapeutic effects are needed. Such experiments are planned but are beyond the scope of this paper, which is already very large (7 Figures and 16 Supplemental Figures).

(3) While the human relevance of TM3 cells is suggested through marker overlap, more quantitative approaches, such as cell identity mapping or gene signature scoring in human datasets, would strengthen the translational connection.

We appreciate the reviewer’s suggestion and agree that additional quantitative analyses will further strengthen the translational relevance of TM3 cells. It is not yet clear if humans have a direct TM3 counterpart or if TM cell roles are compartmentalized differently between human cell types. We are currently limited in our ability to perform these comparative analyses. Specifically, we were unable to obtain permission to use the underlying dataset from Patel et al., and our access to the Van Zyl et al. dataset was through the Single Cell Portal, which does not support more complex analyses (ex. cell identity mapping or gene signature scoring). Differences between human studies themselves also affect these comparisons. Future work aimed at resolving differences and standardizing human TM cell annotations, as well as cross species comparisons are needed (working groups exist and this ongoing effort supports 3 human TM cell subtypes as also reported by Van Zyl). This is beyond what we are currently able to do for this paper. We present a comprehensive assessment using readily available published resources.

**Reviewer #2 (Public review):**
Summary:This elegant study by Tolman and colleagues provides fundamental findings that substantially advance our knowledge of the major cell types within the limbus of the mouse eye, focusing on the aqueous humor outflow pathway. The authors used single-cell and single-nuclei RNAseq to very clearly identify 3 subtypes of the trabecular meshwork (TM) cells in the mouse eye, with each subtype having unique markers and proposed functions. The U. Columbia results are strengthened by an independent replication in a different mouse strain at a separate laboratory (Duke). Bioinformatics analyses of these expression data were used to identify cellular compartments, molecular functions, and biological processes. Although there were some common pathways among the 3 subtypes of TM cells (e.g., ECM metabolism), there also were distinct functions. For example:TM1 cell expression supports heavy engagement in ECM metabolism and structure, as well as TGFb2 signaling.TM2 cells were enriched in laminin and pathways involved in phagocytosis, lysosomal function, and antigen expression, as well as End3/VEGF/angiopoietin signaling.TM3 cells were enriched in actin binding and mitochondrial metabolism.They used high-resolution immunostaining and in situ hybridization to show that these 3 TM subtypes express distinct markers and occupy distinct locations within the TM tissue. The authors compared their expression data with other published scRNAseq studies of the mouse as well as the human aqueous outflow pathway. They used ATAC-seq to map open chromatin regions in order to predict transcription factor binding sites. Their results were also evaluated in the context of human IOP and glaucoma risk alleles from published GWAS data, with interesting and meaningful correlations. Although not discussed in their manuscript, their expression data support other signaling pathways/ proteins/ genes that have been implicated in glaucoma, including: TGFb2, BMP signaling (including involvement of ID proteins), MYOC, actin cytoskeleton (CLANs), WNT signaling, etc.In addition to these very impressive data, the authors used scRNAseq to examine changes in TM cell gene expression in the mouse glaucoma model of mutant Lmxb1-induced ocular hypertension. In man, LMX1B is associated with Nail-Patella syndrome, which can include the development of glaucoma, demonstrating the clinical relevance of this mouse model. Among the gene expression changes detected, TM3 cells had altered expression of genes associated with mitochondrial metabolism. The authors used their previous experience using nicotinamide to metabolically protect DBA2/J mice from glaucomatous damage, and they hypothesized that nicotinamide supplementation of mutant Lmx1b mice would help restore normal mitochondrial metabolism in the TM and prevent Lmx1b-mediated ocular hypertension. Adding nicotinamide to the drinking water significantly prevented Lmxb1 mutant mice from developing high intraocular pressure. This is a laudable example of dissecting the molecular pathogenic mechanisms responsible for a disease (glaucoma) and then discovering and testing a potential therapy that directly intervenes in the disease process and thereby protects from the disease.Strengths:There are numerous strengths in this comprehensive study including:Deep scRNA sequencing that was confirmed by an independent dataset in another mouse strain at another university.Identification and validation of molecular markers for each mouse TM cell subset along with localization of these subsets within the mouse aqueous outflow pathway.Rigorous bioinformatics analysis of these data as well as comparison of the current data with previously published mouse and human scRNAseq data.Correlating their current data with GWAS glaucoma and IOP "hits".Discovering gene expression changes in the 3 TM subgroups in the mouse mutant Lmx1b model of glaucoma.Further pursuing the indication of dysfunctional mitochondrial metabolism in TM3 cells from Lmx1b mutant mice to test the efficacy of dietary supplementation with nicotinamide. The authors nicely demonstrate the disease modifying efficacy of nicotinamide in preventing IOP elevation in these Lmx1b mutant mice, preventing the development of glaucoma. These results have clinical implications for new glaucoma therapies.

We thank the reviewer for these generous and thoughtful comments on the strengths of this study.

Weaknesses:(1) Occasional over-interpretation of data. The authors have used changes in gene expression (RNAseq) to implicate functions and signaling pathways. For example: they have not directly measured "changes in metabolism", "mitochondrial dysfunction" or "activity of Lmx1b".

We thank the reviewer for this feedback. We did not intend to overstate and agree. Our gene expression changes support, but do not by themselves prove, metabolic disturbances. We had felt that this was obvious and did not want to clutter the text. We have revised the manuscript to clarify that our conclusions about metabolic changes and LMX1B activity are based on gene expression patterns rather than direct functional assays and have added EM data (see below under “Recommendations for the authors”).

We have also added the following to the results:

Lines 715-721: “Although the documented gene expression changes strongly suggest metabolic and mitochondrial dysfunction, they do not directly prove it. Using electron microscopy to directly evaluate mitochondria in the TM, we found a reduction in total mitochondria number per cell in mutants (P = 0.015, Figure 6G). In addition, mitochondria in mutants had increased area and reduced cristae (inner membrane folds) in mutants consistent with mitochondrial swelling and metabolic dysfunction (all P < 0.001 compared to WT, Figure 6G-H).”

More detailed EM and metabolic studies are underway but are beyond the scope of this paper.

(2) In their very thorough data set, there is enrichment of or changes in gene expression that support other pathways that have been previously reported to be associated with glaucoma such as TGFb2, BMP signaling, actin cytoskeletal organization (CLANs), WNT signaling, ossification, etc. that appears to be a lost opportunity to further enhance the significance of this work.

We appreciate the reviewer’s suggestions for enhancing the relevance of our work, we had not initially discussed this due to length concerns. We have now incorporated some of this information into the manuscript (see below under “Recommendations for the authors”).

**Reviewer #3 (Public review):**
Summary: In this study, the authors perform multimodal single-cell transcriptomic and epigenomic profiling of 9,394 mouse TM cells, identifying three transcriptionally distinct TM subtypes with validated molecular signatures. TM1 cells are enriched for extracellular matrix genes, TM2 for secreted ligands supporting Schlemm's canal, and TM3 for contractile and mitochondrial/metabolic functions. The transcription factor LMX1B, previously linked to glaucoma, shows the highest expression in TM3 cells and appears to regulate mitochondrial pathways. In Lmx1bV265D mutant mice, TM3 cells exhibit transcriptional signs of mitochondrial dysfunction associated with elevated IOP. Notably, vitamin B3 treatment significantly mitigates IOP elevation, suggesting a potential therapeutic avenue.This is an excellent and collaborative study involving investigators from two institutions, offering the most detailed single-cell transcriptomic and epigenetic profiling of the mouse limbal tissues-including both TM and Schlemm's canal (SC), from wild-type and Lmx1bV265D mutant mice. The study defines three TM subtypes and characterizes their distinct molecular signatures, associated pathways, and transcriptional regulators. The authors also compare their dataset with previously published murine and human studies, including those by Van Zyl et al., providing valuable crossspecies insights.Strengths:(1) Comprehensive dataset with high single-cell resolution(2) Use of multiple bioinformatic and cross-comparative approaches(3) Integration of 3D imaging of TM and SC for anatomical context(4) Convincing identification and validation of three TM subtypes using molecular markers.

We thank the reviewer for their comments on the strengths of this study.

Weaknesses:(1) Insufficient evidence linking mitochondrial dysfunction to TM3 cells in Lmx1bV265D mice: While the identification of TM3 cells as metabolically specialized and Lmx1b-enriched is compelling, the proposed link between Lmx1b mutation and mitochondrial dysfunction remains underdeveloped. It is unclear whether mitochondrial defects are a primary consequence of Lmx1b-mediated transcriptional dysregulation or a secondary response to elevated IOP. Additional evidence is needed to clarify whether Lmx1b directly regulates mitochondrial genes (e.g., via ChIP-seq, motif analysis, or ATAC-seq), or whether mitochondrial changes are downstream effects.

We agree and refer the reviewer to our responses to the other referees including Reviewer 1, Comment 1 and Reviewer 2 comments 1 and 17. As noted there, these mechanistic questions are the focus of ongoing and future studies. We have revised the text where appropriate to ensure it accurately reflects the scope of our current data.

(2) Furthermore, the protective effects of nicotinamide (NAM) are interpreted as evidence of mitochondrial involvement, but no direct mitochondrial measurements (e.g., immunostaining, electron microscopy, OCR assays) are provided. It is essential to validate mitochondrial dysfunction in TM3 cells using in vivo functional assays to support the central conclusion of the paper. Without this, the claim that mitochondrial dysfunction drives IOP elevation in Lmx1bV265D mice remains speculative. Alternatively, authors should consider revising their claims that mitochondrial dysfunction in these mice is a central driver of TM dysfunction.

We again refer the reviewer to our other response including Reviewer 1, Comment 1 and Reviewer 2 comments 1 and 17.

(3) Mechanism of NAM-mediated protection is unclear: The manuscript states that NAM treatment prevents IOP elevation in Lmx1bV265D mice via metabolic support, yet no data are shown to confirm that NAM specifically rescues mitochondrial function. Do NAM-treated TM3 cells show improved mitochondrial integrity? Are reactive oxygen species (ROS) reduced? Does NAM also protect RGCs from glaucomatous damage? Addressing these points would clarify whether the therapeutic effects of NAM are indeed mitochondrial.

We refer the reviewer to our response to Reviewer 1, Comment 2.

(4) Lack of direct evidence that LMX1B regulates mitochondrial genes: While transcriptomic and motif accessibility analyses suggest that LMX1B is enriched in TM3 cells and may influence mitochondrial function, no mechanistic data are provided to demonstrate direct regulation of mitochondrial genes. Including ChIP-seq data, motif enrichment at mitochondrial gene loci, or perturbation studies (e.g., Lmx1b knockout or overexpression in TM3 cells) would greatly strengthen this central claim.

We refer the reviewer to our response to Reviewer 1, Comment 1.

(5) Focus on LMX1B in Fig. 5F lacks broader context: Figure 5F shows that several transcription factors (TFs)-including Tcf21, Foxs1, Arid3b, Myc, Gli2, Patz1, Plag1, Npas2, Nr1h4, and Nfatc2exhibit stronger positive correlations or motif accessibility changes than LMX1B. Yet the manuscript focuses almost exclusively on LMX1B. The rationale for this focus should be clarified, especially given LMX1B's relatively lower ranking in the correlation analysis. Were the functions of these other highly ranked TFs examined or considered in the context of TM biology or glaucoma? Discussing their potential roles would enhance the interpretation of the transcriptional regulatory landscape and demonstrate the broader relevance of the findings.

Our analysis (Figure 5F) indicates that Lmx1b is the transcription factor most strongly associated with its predicted target gene expression across all TM cells, as reflected by its highest value along the X-axis. While other transcription factors exhibit greater motif accessibility (Y-axis), this likely reflects their broader expression across TM subtypes. In contrast, Lmx1b is minimally expressed in TM1 and TM2 cells, which may account for its lower motif accessibility overall (motifs not accessible in cells where Lmx1b is not / minimally expressed).

Our emphasis on LMX1B is further supported by its direct genetic association with glaucoma. In contrast, the other transcription factors lack clear links to glaucoma and are supported primarily by indirect evidence. Nonetheless, we agree that the transcription factors highlighted in our analysis are promising candidates for future investigation. However, to maintain focus on the central narrative of this study, we have chosen not to include an extended discussion of these additional genes.

(6) In abstract, they say a number of 9,394 wild-type TM cell transcriptomes. The number of Lmx1bV265D/+ TM cell transcriptomes analyzed is not provided. This information is essential for evaluating the comparative analysis and should be clearly stated in the Abstract and again in the main text (e.g., lines 121-123). Including both wild-type and mutant cell counts will help readers assess the balance and robustness of the dataset.

We thank the reviewer for noticing this oversight and have added this value to the abstract and results section.

Lines 41 and 696: 2,491 mutant TM cells.

(7) Did the authors monitor mouse weight or other health parameters to assess potential systemic effects of treatment? It is known that the taste of compounds in drinking water can alter fluid or food intake, which may influence general health. Also, does Lmx1bV265D/+ have mice exhibit non-ocular phenotypes, and if so, does nicotinamide confer protection in those tissues as well? Additionally, starting the dose of the nicotinamide at postnatal day 2, how long the mice were treated with water containing nicotinamide, and after how many days or weeks IOP was reduced, and how long the decrease in the IOP was sustained.

Water intake was monitored in both treatment groups, and dosing was based on the average volume consumed by adult mice (lines 1017–1018, young pups do not drink water and so drug is largely delivered through mothers’ milk until weaning and so we do not know an accurate dose for young pups). Mouse health was assessed throughout the experiment through regular monitoring of body weight and general condition.

Depending on genetic context, Lmx1b mutations can cause kidney disease and impact other systems. Non-ocular phenotypes were not the focus of this study and were not characterized.

We added a comment to the method to clarify the NAM treatment timeline. NAM was administered continuously in the drinking water starting at P2 and maintained throughout the experiment. IOP was measured beginning at 2 months and then at monthly time points. NAM lessened IOP at 2 and 3 months. We terminated IOP assessment at 3 months.

Lines 1028-1029: “Treatment was started at postnatal day 2 and continued throughout the experiment.”

(8) While the IOP reduction observed in NAM-treated Lmx1bV265D/+ mice appears statistically significant, it is unclear whether this reflects meaningful biological protection. Several untreated mice exhibit very high IOP values, which may skew the analysis. The authors should report the mean values for IOP in both untreated and NAM-treated groups to clarify the magnitude and variability of the response.

We have added supplemental table 7 with the statistical information. Regarding the high IOP values observed in a subset of untreated V265D mutant mice, we consistently detect individual mutant eyes with IOPs exceeding 30 mmHg across independent cohorts and time points [3-5]. It is important to note that IOP is subject to fluctuation and in disease states such as glaucoma, circadian rhythms can be disrupted with stochastic and episodic IOP spikes throughout the day. This may be occurring in those untreated mice. This is also why we strive to use sample sizes of 40 or more. Additionally, we observe that some mutant eyes with IOPs measured within the normal range have anterior chamber deepening (ACD) - a persistent anatomical change associated with sustained or recurrent high IOP that stretches the cornea and may posteriorly displace the lens. This suggests mutant mice experience transient IOP elevations that are not always captured at a single time point due to the stochastic nature of these fluctuations. To account for this, we include ACD as an additional readout alongside IOP measurements. The reduction in ACD observed in NAM-treated mice provides independent evidence supporting the biological relevance of NAM-mediated IOP reduction.

(9) Additionally, since NAM has been shown to protect RGCs in other glaucoma models directly, the authors should assess whether RGCs are preserved in NAM-treated Lmx1b V265D/+ mice. Demonstrating RGC protection would support a synergistic effect of NAM through both IOP reduction and direct neuroprotection, strengthening the translational relevance of the treatment.

We again thank the referee. We note the possibility of dual IOP protection and neuroprotection in the manuscript (lines 961–963). The goal of the present study, however, was to determine mechanisms underlying IOP elevation in patients with LMX1B variants. Therefore, we limited our focus to IOP elevation (LMX1B is expressed in the TM but not RGCs). Studies of the RGCs and optic nerve in V265D mutant mice treated with NAM take considerable effort but are underway. They will be reported in a subsequent manuscript. Initial data support protection, but that is a work in progress.

Additionally, we recently reported a similar pattern of IOP protection to that reported here using pyruvate - in experiments where we analyzed the optic nerve as the focus of the study was assessment of pyruvate as a resilience factor against high genetic risk of glaucoma [4]. In that case, there was statistically significant protection from glaucomatous optic nerve damage, arguing for translational relevance again with a possible synergistic effect through both IOP reduction and direct neuroprotection.

(10) Can the authors add any other functional validation studies to explore to understand the pathways enriched in all the subtypes of TM1, TM2, and TM3 cells, in addition to the ICH/IF/RNAscope validation?

We agree with the reviewer on the importance of further functional validation of pathways active in TM cell subtypes that influence IOP. However, comprehensive investigation of the pathways active in subtypes need to be in future studies. It is beyond the scope of his already large paper.

(11) The authors should include a representative image of the limbal dissection. While Figure S1 provides a schematic, mouse eyes are very small, and dissecting unfixed limbal tissue is technically challenging. It is also difficult to reconcile the claim that the majority of cells in the limbal region are TM and endothelium. As shown in Figure S6, DAPI staining suggests a much higher abundance of scleral cells compared to TM cells within the limbal strip. Additional clarification or visual evidence would help validate the dissection strategy and cellular composition of the captured region.

We appreciate the reviewer’s suggestion and have added additional images to Figure S1 to show our limbal strip dissection. However, we clarify that we do not intend to suggest that TM and endothelial cells are the most abundant populations in these dissected strips. When we say “are enriched for drainage tissues” we mean in comparison to dissecting the anterior segment as a whole. We have clarified this in the text. In fact, epithelial cells (primarily from the cornea) constituted the largest cluster in our dataset (Figure 1A). Additionally, to avoid misinterpretation, we generally refrain from drawing conclusions about the relative abundance of cell types based on sequencing data. Single-cell and single nucleus RNA sequencing results are sensitive to technical factors that alter cell proportions depending on exact methodological details. In our study, TM cells comprised 24.4% of the single-cell dataset and 11.8% of the single-nucleus dataset, illustrating the impact of methodological variability.

Lines 163-164: “Individual eyes were dissected to isolate a strip of limbal tissue, which is enriched for TM cells in comparison to dissecting the anterior segment as a whole.”

**Reviewer #1 (Recommendations for the authors):**
To enhance the reproducibility and transparency of the findings presented in this study, we strongly recommend that the authors make all analysis scripts and computational tools publicly available.

We agree with the reviewer’s emphasis on transparency and are currently building a GitHub page to share our scripts. However, we did not develop any new tools for this study. All tools that we used are publicly available and provided in our methods section. All data will be available as raw data and through the Broad Institute’s Single Cell Portal.

**Reviewer #2 (Recommendations for the authors):**
The authors are to be commended for a well-written presentation of high-quality data, their comparisons of datasets (other mouse and human scRNAseq data), correlation with clinical glaucoma risk alleles, and curative therapy for the mouse model of Lmx1b glaucoma. There are several minor suggestions that the authors might consider to further improve their manuscript:(1) Lines 42-43: Although their data strongly support the role of mitochondrial dysfunction in Lmx1b glaucoma, they might want to soften their conclusion "supports a primary role of mitochondrial dysfunction within TM3 cells initiating the IOP elevation that causes glaucoma".

With the inclusion of EM data supporting mitochondrial dysfunction in Lmx1b mutant TM cells, we have revised this sentence to more accurately reflect our findings.

Lines 42-44 (previously lines 42-43): “Mitochondria in TM cells of V265D/+ mice are swollen with a reduced cristae area, further supporting a role for mitochondrial dysfunction in the initiation of IOP elevation in these mice.”

(2) Figure 1: Why is the shape of the "TM containing" cluster in 1A so different than the cluster shown in 1B?

We isolated cells from the 'TM-containing' cluster and performed unbiased reclustering, which alters their positioning in UMAP space. The figure legend has been updated to clarify this point.

Lines 143-144 “A separate UMAP representation of the trabecular meshwork (TM) containing cluster following subclustering.”

(3) Line 160: change "data was" to "data were"

Corrected

(4) S4 Fig C: Please comment on why the Columbia and Duke heatmaps for TM3 are not as congruent as the heatmaps for TM1 and TM2.

We cannot definitively determine the reason for this. However, differences in tissue processing techniques between the Columbia and Duke preparations may contribute. Such variations have been shown to affect cellular transcriptomes in certain contexts. It is possible that TM3 cells are more susceptible to these effects than others. We have added a statement addressing this point to the figure legend.

Lines 238-240: “Because tissue processing techniques can alter gene expression [52], the heatmap variation between institutes likely reflects differences in processing techniques (Methods) and suggests that TM3 cells are more susceptible to these effects than other cell types.”

(5) S9 Fig: It is very difficult to see any staining for TM1 CHIL1 (2nd panel), TM2 End3 (2nd panel), and TM3 Lypd1 (both panels)

We apologize for the difficulty in visualizing these panels. To improve clarity, we have increased the brightness of all relevant marker signals, within standard bounds, to facilitate easier interpretation.

(6) Line 380: "are significantly higher"; since statistical analysis was not reported, please do not use "significantly"

Done

(7) The authors should consider discussing several of their findings that agree with published literature. For example:Figure 3B: "Wnt protein binding" (PMID: 18274669), "TGFb "binding" (numerous references), "integrin binding" (work of Donna Peters), "actin binding"/"actin filament binding"/"actin filament bundle" (CLANs references)S10 Fig c: "ossification" (work of Torretta Borres)S11 Fig A: ID2/ID3 (PMID: 33938911); (B) BMP4 (PMID: 17325163)S12 Fig A: MYOC in TM1 cells (numerous references)

We appreciate the reviewer’s diligent review and comments regarding these pathways. We have added a comment to the discussion regarding the agreement of these pathways.

Lines 855-858: In addition, the expression of genes that we document generally agrees with the literature. For example, the following genes and signaling molecules have been reported in TM cells, WNT signaling [78], TGF-β signaling [79-85], integrin binding [86-88], actin cytoskeletal networks [89], calcification genes [90, 91], and Myocilin [91-94].

(8) Line 541: was confocal microscopy used to measure the "3D shapes" of nuclei or was this done with a single image to determine sphericity?

This analysis was performed using confocal microscopy and 3D reconstructed models of the TM nuclei. We have added text to clarify this in the figure legend

Lines 553-556: “To rigorously assess whether TM1 nuclei are more spherical, we analyzed their reconstructed 3D shapes from whole mounts images by confocal microscopy, comparing them to TM3 nuclei using the ‘Sphericity’ tool in Imaris.”

(9) Line 545: please add a close parentheses after "scoring 1"

Done

(10) S15 Fig: (A) There does not appear to be "good agreement" (line 653) between the datasets for TM1. (C) please provide a better explanation on how to interpret these "Confusion Matrix" results.

We understand the referee's concern, the patterns likely appear different to the referee due to limited sampling in snRNA-seq data. Based on our results, TM1 seems particularly susceptible, possibly because these cells do not tolerate the isolation process as well. Although we are confident that TM1 shows good agreement between the two techniques based on our experience, we have revised the language in the text to “generally” to reflect this nuance.

Lines 633-635 (previously line 653): The generated clusters and their marker genes generally agreed with our scRNA-seq analyses (Fig 5A-B, S15A Fig).

We have also added additional clarification for how to interpret the Confusion Matrix.

Lines 669-672: “Colors indicate the fraction of cells identified in each ATAC cluster (row) which are also identified in each RNA cell type (columns), where darker colors represent stronger correspondence between RNA and ATAC clusters.”

(11) Line 676: The transition from discussing the sc/snRNAseq data to the work in Lmx1b mutant mice is quite abrupt and could use a better transition to introduce this metabolism work.

We have revised this transition for improved flow but prefer to keep all transitions brief due to the paper's length.

Lines 691-694 (previously line 676): To evaluate the utility of our new TM cell atlas, we used it to examine how Lmx1b mutations affect the TM cell transcriptome and to identify potential mechanisms underlying IOP elevation. We selected LMX1B because it causes IOP elevation and glaucoma in humans and was identified as a highly active transcription factor in our TM cell dataset.

(12) Lines 696-697: It appears counter-intuitive that upregulation of ubiquitin pathways would lead to proteostasis (proteosome protein degradation requires ubiquination).

We have clarified that the protein tagging pathway was significantly upregulated. However, polyubiquitin precursor itself was downregulated. In general, the statistical significance of the protein tagging pathway suggests perturbation of the system tagging proteins for degradation. We have clarified this in the text.

Lines 711-714 (previously lines 696-697): “In addition, mutant TM3 cells showed an upregulation of protein tagging genes. However, there is a downregulation of the polyubiquitin precursor gene (Ubb, P = 4.5E-30), indicating a general dysregulation of pathways that tag proteins for degradation.”

(13) Line 715: Please justify why "perturbed metabolism" was chosen to pursue vs the other differentially expressed pathways

We chose to narrow our focus on TM3 cells because of the enrichment for Lmx1b expression.Most pathways identified in our analysis of TM3 cells implicate mitochondrial metabolism.Therefore, we chose to further explore this avenue. We clarified that perturbed metabolism was the strongest gene expression signature in the text.

Lines 753-754 (previously line 715): “Our findings most strongly implicate perturbed metabolism within TM3 cells as responsible for IOP elevation in an Lmx1b glaucoma model.”

(14) Line 759: The authors clearly demonstrate that Lmx1b is most expressed in TM3 cells; however, they did not demonstrate that "Lmx1b was most active"

ATAC analysis showed that Lmx1b was most active in TM cells overall. We inferred its activity in TM3 because Lmx1b is most enriched in that subtype. This has been clarified in the text.

Lines 799-800 (previously line 759): “More specifically, we demonstrate that Lmx1b is the most active TM cell TF and is enriched in TM3 cells,…”

(15) Lines 830-835: Please include references documenting increased TGFβ2 concentrations in POAG aqueous humor and TM, effects of TGFβ2 on TM ECM deposition, and TGFβ2 induced ocular hypertension ex vivo and in vivo.

Done.

(16) Line 875: The authors provide no direct evidence for enhances "oxidative stress" in Lmx1b TM3 cells

The mitochondrial abnormalities and changed pathways support oxidative stress, but we have not directly tested this. Experiments are currently underway to evaluate its role, but these additional analyses are beyond the scope of this paper. We removed oxidative stress from the sentence.

Lines 920-922 (previously line 875): “Importantly, in heterozygous mutant V265D/+ mice, TM3 cells had pronounced gene expression changes that implicate mitochondrial dysfunction, but that were absent or much lower in other cells including TM1 and TM2.”

(17) Line 880: Similarly, the authors have not directly assessed effects on metabolism in TM3 cells; they only have shown changes in the expression of mitochondrial genes that may affect metabolism

We have no way to specifically isolating TM3 cells to test this. Future work is underway to test this more broadly in isolated TM cells but is beyond the scope of this is already large paper. Considering our gene expression data and the addition of supporting EM data, we have qualified the text.

Lines 930-931 (previously 880): “Our data extend these published findings by showing that inheritance of a single dominant mutation in Lmx1b similarly affects mitochondria in TM cells.”

(18) Line 892: What markers were used to detect "cell stress"?

We have revised the text. Although our RNA data show stress gene changes, characterization of these markers is beyond the scope of the current study and will be included in a subsequent paper.

Lines 945-948 (previously line 892): “However, these processes were not limited to TM3 cells or even to cell types that express detectable Lmx1b, suggesting that they are secondary damaging processes that are subsequent to the initiating, Lmx1b-induced perturbations in TM3 cells.”

**Additional author driven change**

While revising and reviewing our data, we identified a coding error that resulted in the WT and V265D mutant group labels being switched in Figure 6. Importantly, the significance of the differentially expressed genes (DEGs), the implicated biological pathways, and the interpretation of pathway directionality in the manuscript remain accurate. The only issue was the incorrect labeling in the figure. We have corrected the labels in Figure 6 to accurately reflect the data. As noted above, all data and code will be made available to ensure full reproducibility of our results.

References

(1) Doucet-Beaupre H, Gilbert C, Profes MS, Chabrat A, Pacelli C, Giguere N, et al. Lmx1a and Lmx1b regulate mitochondrial functions and survival of adult midbrain dopaminergic neurons. Proc Natl Acad Sci U S A. 2016;113(30):E4387-96. Epub 2016/07/14. doi: 10.1073/pnas.1520387113. PubMed PMID: 27407143; PubMed Central PMCID: PMCPMC4968767.

(2) Jimenez-Moreno N, Kollareddy M, Stathakos P, Moss JJ, Anton Z, Shoemark DK, et al. ATG8-dependent LMX1B-autophagy crosstalk shapes human midbrain dopaminergic neuronal resilience. J Cell Biol. 2023;222(5). Epub 2023/04/05. doi: 10.1083/jcb.201910133. PubMed PMID: 37014324; PubMed Central PMCID: PMCPMC10075225.

(3) Cross SH, Macalinao DG, McKie L, Rose L, Kearney AL, Rainger J, et al. A dominantnegative mutation of mouse Lmx1b causes glaucoma and is semi-lethal via LDB1mediated dimerization [corrected]. PLoS Genet. 2014;10(5):e1004359. Epub 2014/05/09. doi: 10.1371/journal.pgen.1004359. PubMed PMID: 24809698; PubMed Central PMCID: PMCPMC4014447.

(4) Li K, Tolman N, Segre AV, Stuart KV, Zeleznik OA, Vallabh NA, et al. Pyruvate and related energetic metabolites modulate resilience against high genetic risk for glaucoma. Elife. 2025;14. Epub 2025/04/24. doi: 10.7554/eLife.105576. PubMed PMID: 40272416; PubMed Central PMCID: PMCPMC12021409.

(5) Tolman NG, Balasubramanian R, Macalinao DG, Kearney AL, MacNicoll KH, Montgomery CL, et al. Genetic background modifies vulnerability to glaucoma-related phenotypes in Lmx1b mutant mice. Dis Model Mech. 2021;14(2). Epub 2021/01/20. doi: 10.1242/dmm.046953. PubMed PMID: 33462143; PubMed Central PMCID: PMCPMC7903917.